# Learning Generalized Linear Programming Value Functions

**Tu Anh-Nguyen**
Google Research and Rice University
Houston, TX
tu.na@rice.edu

**Joey Huchette**
Google Research
Cambridge, MA
jhuchette@google.com

**Christian Tjandraatmadja**
Google Research
Cambridge, MA
ctjandra@google.com

## Abstract

We develop a theoretically-grounded learning method for the *Generalized Linear Programming Value Function* (GVF), which models the optimal value of a linear programming (LP) problem as its objective and constraint bounds vary. This function plays a fundamental role in algorithmic techniques for large-scale optimization, particularly in decomposition for two-stage mixed-integer linear programs (MILPs). This paper establishes a structural characterization of the GVF that enables it to be modeled as a particular neural network architecture, which we then use to learn the GVF in a way that benefits from three notable properties. First, our method produces a true under-approximation of the value function with respect to the constraint bounds. Second, the model is input-convex in the constraint bounds, which not only matches the structure of the GVF but also enables the trained model to be efficiently optimized over using LP. Finally, our learning method is unsupervised, meaning that training data generation does not require computing LP optimal values, which can be prohibitively expensive at large scales. We numerically show that our method can approximate the GVF well, even when compared to supervised methods that collect training data by solving an LP for each data point. Furthermore, as an application of our framework, we develop a fast heuristic method for large-scale two-stage MILPs with continuous second-stage variables, via a compact reformulation that can be solved faster than the full model linear relaxation at large scales and orders of magnitude faster than the original model.

## 1 Introduction

The *linear programming (LP) value function* models the optimal value of an LP as problem data in that problem varies. Value functions are a fundamental abstraction used in many algorithms for large-scale optimization. More concretely, many problems where decisions are made sequentially–e.g., two-stage stochastic programs [46], facility location problem [17], multi-commodity problems [19], or network interdiction problems [49]–can be modeled as two-stage mixed-integer linear programs (MILPs). To pick one common technique as a motivating example: *Benders' decomposition* is an algorithmic technique that decomposes such a large problem into many smaller ones [21, 42, 51]. At a high level, Benders' decomposition abstracts each LP subproblem away by replacing it with a value function. If we somehow had a good approximation of these value functions that we could efficiently optimize over, this reformulation would be straightforward to approximately solve. In Benders' decomposition however, we do not have such a representation *a priori*, and thus we iteratively

38th Conference on Neural Information Processing Systems (NeurIPS 2024).

construct an approximation of it via cutting planes; this is often the most computationally expensive part of the algorithm [36, 57]. In this paper, we focus on learning such representations.

Separately, it is well-known that neural networks (NNs) are "universal approximators" in theory [4, 23, 26, 35] and incredibly adept at modeling complex behaviors in practice [1, 7, 9, 37]. Taken together, we can state two natural questions that motivate this work:

- What are the meaningful structural properties of a value function, and what are suitable neural network architectures for encoding these properties?
- How good are these approximations in practice, and how can we leverage them to solve real-world problems?

**Contributions.** Our work studies the *Generalized Linear Programming Value Function* (GVF), defined as the function that models an LP's optimal value as both its objective and its constraint bounds vary, and shows how machine learning techniques can be used to build practical approximations of this function. In particular, our contributions are as follows.

1. **A GVF Representation theorem.** We study the structure of the GVF and show that it can be exactly modeled as a maximum of bilinear functions, where each function is the dot product of two piecewise linear functions that depends only the objective coefficients or constraint bounds, respectively.

2. **A theoretically-grounded NN architecture for GVF.** We present the *Dual-Stack Model*, a neural network architecture which mimics the structural property of the GVF exposed by our representation theorem.

3. **An unsupervised learning approach.** We show that the GVF can be written as the unique optimal solution of a constrained optimization problem that does not require solving any LPs to write down. We use this as inspiration for an unsupervised learning method that can be implemented using standard NN training constructs and libraries.

4. **Empirical justification.** We present a computational study showing that our unsupervised training approach can perform comparably with supervised training in terms of approximating a GVF, without the expensive data generation phase that supervised training requires.

5. **A fast heuristic for large-scale two-stage MILPs.** Due to the properties of the Dual-Stack Model, we can easily embed it as an LP within a larger optimization problem. As an application of our framework, we leverage this fact to produce a heuristic for two-stage MILPs with continuous second-stage variables, which includes a provable duality gap.

## 2 Preliminaries

A Linear Program (LP) is a mathematical optimization problem of the form:

$$\min\{c \cdot x \mid Ax \leq b, x \geq 0, x \in \mathbb{R}^n\}, \tag{1}$$

where $x$ are the decision variables, $c \in \mathbb{R}^n$ is the vector of objective coefficients, $A \in \mathbb{R}^{m \times n}$ is the constraint matrix, and $b \in \mathbb{R}^m$ is the vector of constraint bounds (often called the "right-hand side" in such a representation).

Duality is a fundamental concept in linear programming that establishes a relationship between the primal (original) linear program and its dual (related) linear program. This relationship provides insights into the optimal solutions, and it is valuable for both theoretical understanding and practical applications. The dual problem of (1) is

$$\max\{b \cdot y \mid A^T y \leq c, y \leq 0, y \in \mathbb{R}^m\}. \tag{2}$$

The typical LP Value Function (LPVF) is $h_{A,c}(\beta) := \min_x \{c \cdot x \mid Ax \leq \beta, x \geq 0, x \in \mathbb{R}^n\}$. As a corollary of strong duality, the LPVF is a piecewise linear convex function, which is the maximum of a finite number of affine functions. Note that the LPVF only considers varying constraint bounds; if we also permit the objective coefficients to vary, we obtain the Generalized LP Value Function (GVF) [52]. Formally, we define a GVF associated with a fixed constraint matrix $A \in \mathbb{R}^{m \times n}$ as:

$$h_A(\gamma, \beta) := \min_x \{ \gamma \cdot x \mid Ax \leq \beta, x \geq 0 \}. \tag{3}$$

Many typical decomposition methods only need to consider LPVFs of fixed objective vectors $c$. However, learning the entire GVF at once means that we can reuse the same learned model for many different objectives, potentially saving computation and allowing for a broader generalization.

We use $LP(\gamma, \beta)$ to denote the linear program in (3) for fixed values of $\gamma$ and $\beta$. Conventionally, when $LP(\gamma, \beta)$ is infeasible, $h(\gamma, \beta) = +\infty$, and when $LP(\gamma, \beta)$ is unbounded from below, we have $h(\gamma, \beta) = -\infty$. Let $\mathcal{B} := \{\beta \in \mathbb{R}^m \mid \exists x \in \mathbb{R}^n \text{ s.t } Ax \leq \beta, x \geq 0\}$ and $\mathcal{C} := \{\gamma \in \mathbb{R}^n \mid \exists y \in \mathbb{R}^m \text{ s.t } A^T y \leq \gamma, y \leq 0\}$. By strong duality and the definitions of $\mathcal{B}$ and $\mathcal{C}$: $h(\gamma, \beta)$ is finite if and only if $\gamma \in \mathcal{C}$ and $\beta \in \mathcal{B}$. We define $X(\beta) := \{x \in \mathbb{R}^m \mid Ax \leq \beta, x \geq 0\}$ as the set of feasible solutions of $LP(\cdot, \beta)$ for a fixed $\beta \in \mathcal{B}$.

In this work, we will consider two-stage MILPs with continuous second-stage variables, where fixing the first-stage variables results in the problem decomposing into $K$ independent LPs. In particular, these have the form:

$$\min_{x^1, x^2} \left\{ c \cdot x^1 + \sum_{k \in [\![K]\!]} d^k \cdot x^{2,k} \mid x^1 \in \chi, \ T^k x^1 + A x^{2,k} \leq b^k, \ x^{2,k} \geq 0 \ \forall k \in [\![K]\!] \right\}, \quad (4)$$

where $x^1 \in \mathbb{R}^{n_1}$ are the first-stage variables, $\chi$ is the first-stage feasible set, $K \in \mathbb{Z}^+$ is the number of second-stage subproblems, and $[\![K]\!]$ denotes the set $\{1, \ldots, K\}$. For example, in the context of stochastic programming, $K$ is the number of scenarios, or in a facility location problem, $K$ is the number of customers. Each subproblem $k$ is associated with corresponding continuous second-stage variables $x^{2,k} \in \mathbb{R}^{n_2}$. We assume that the second-stage constraint matrices are the same, denoted as $A \in \mathbb{R}^{m_2 \times n_2}$ while the constraint matrices of first-stage and the constraint bounds constraints vector can vary among second-stages, denoted as $T^k$ and $b^k$, respectively. We can rewrite (4) using GVF as

$$\min_{x^1} \left\{ c \cdot x^1 + \sum_{k \in [\![K]\!]} h_A(d^k, b^k - T^k x^1) \mid x^1 \in X \right\}. \quad (5)$$

Within the context of GVFs, we can reformulate equation (3) to elide the requirement for second-stage variables. The large number of these variables can impede computational efficiency [18], which motivates a compact representation of GVFs.

## 3 Related Work

### 3.1 Value function learning for multi-stage problems

Neural networks are well-known to be powerful "universal approximators" [35]. This has motivated a line of research focused on learning value functions, particularly those based on constraint bounds such as LPVF and their MILP analogues, with the main goal of improving methods for two-stage or multi-stage optimization problems. Dai et al. [13], Lee et al. [34], and Bae et al. [5] propose various methods to learn the LPVF with the aim to solve multi-stage stochastic programming problems more quickly. Similar to our work, they use models that are convex on constraint bounds to match the structure of the LPVF. Beyond the LP value function, neural networks have also been used to learn IP value functions to improve the integer L-shaped method [33]. Moreover, to tackle difficult mixed-integer problems with a large number of scenarios, Dumouchelle et al. [14, 15] devise NN architectures that learn MILP value functions of constraint bounds and scenarios.

Our method differs from the above in that we directly learn the GVF rather than the LPVF (though [14] learns the GVF indirectly), we do not require solving optimization subproblems (LPs in our case) to obtain training data. Furthermore, we aim to not only generalize across second-stage subproblems (scenarios), but also across instances. In particular, by allowing the objective coefficients to vary, we learn a single value function that encompasses all subproblems, rather than learning one per subproblem. Of course, learning a single GVF is generally harder than learning a single LPVF, but a core thesis of this work is that there is underlying structure tying together those many related LPVFs that we can exploit when learning the GVF.

### 3.2 Other learning-based approaches

A related research direction in learning for stochastic optimization is scenario reduction, which seeks a smaller set of "representative scenarios". Many of these approaches perform some form

of clustering to reduce the number of scenarios and then solve a smaller surrogate problem with these scenarios [10, 16, 30, 41, 44]. Wu et al. [55] uses a conditional variational autoencoder to learn scenario embeddings and cluster them. Bengio et al. [8] predicts a representative scenario for a smaller surrogate problem, but it relies on problem structure to build scenarios for training.

Other learning-based methods for tackling two-stage stochastic problems include reinforcement learning for local search [38] and Benders cut classification [28]. More generally, ML-based approaches have also been applied for bilevel optimization [29, 47, 48, 56] where value functions are relevant, though unlike in our case, these only have a single inner optimization subproblem. Finally, there is an extensive stream of work focusing on applying ML to support decisions within MILP solvers, such as branching and cutting plane decisions (e.g., [2, 24, 39, 50]).

### 3.3 Computing value functions

LP value functions are well-studied (e.g., see [45, Chapter 19]), as they play crucial roles in sensitivity analysis and Benders' decomposition. On the other hand, GVFs are considerably less well-studied. While much is known about its structure [22, 27], to the best of our knowledge our method is the first that aims to learn it directly based on its theoretical properties. The computation of value functions has also been studied for ILPs and MILPs using superadditive duality [32, 43, 52, 53]. However, they are less tractable to compute and thus more difficult to leverage into a practical algorithm.

## 4 A Neural Network Representation for Generalized Linear Programming Value Functions

In this section, we will develop a characterization of the GVF that, in the sequel, we will use as inspiration for a neural network architecture that is well-suited to approximate the GVF.

### 4.1 A Characterization of Generalized Linear Programming Value Function

It is known that $\mathcal{B} \times \mathcal{C}$ can be partitioned into distinct *invariancy regions*, within each of which the GVF is bilinear with respect to $\beta$ and $\gamma$ [27]. This can be reformulated as the following proposition.

**Proposition 1.** *Fix a matrix $A \in \mathbb{R}^{m \times n}$, and define $S(\beta)$ as the set of all bases of $A$ which are feasible with respect to fixed constraint bounds $\beta \in \mathcal{B}$. Then, $h_A(\gamma, \beta) = \min_{B \in S(\beta)} \gamma_B B^{-1} \beta$. Furthermore, $h_A(\cdot, b)$ is piecewise linear concave for every fixed $b \in \mathcal{B}$ and $h_A(c, \cdot)$ is piecewise linear convex for every fixed $c \in \mathcal{C}$.*

While Proposition 1 tells us that each invariancy region defined by $\mathcal{B}$ can be decomposed into a product of functions that depend only on $\gamma$ and $\beta$, it does not provide us with a *global* decomposition that is valid across all invariancy regions. We now show that there does indeed exist a structured, global decomposition of a GVF in terms of piecewise linear functions that consider either $\gamma$ or $\beta$, but not both.

**Theorem 2.** *(GVF Representation Theorem) For a fixed matrix $A \in \mathbb{R}^{m \times n}$, there exists a set of $p$ piecewise linear functions $\{F_p : \mathbb{R}^n \to \mathbb{R}^K\}_{p=1}^P$ and a piecewise linear convex function $G : \mathbb{R}^m \to \mathbb{R}^K$ such that*

$$h_A(\gamma, \beta) = \max_{p \in [\![P]\!]} \{F_p(\gamma)^T G(\beta)\} \quad \forall \gamma \in \mathcal{C}, \beta \in \mathcal{B}. \tag{6}$$

We refer the reader to Appendix A for a proof of this result.

### 4.2 The Dual-Stack Model

We now use Theorem 2 as inspiration for a neural network architecture that we dub the *Dual-Stack Model* (DSM). For simplicity, in the remainder we will consider LPs written in the form of (1); see Appendix F for analogous models for other LP representations.

The architecture of a DSM is depicted in Figure 1. It consists of two stacks of feedforward fully-connected neural networks of depth $N$ and $M$ corresponding to the objective vector $\gamma$ and the constraint bounds vector $\beta$; we name them the $\gamma$-stack and the $\beta$-stack, respectively. Each layer

has a piecewise linear activation function to ensure that the entire stack itself is piecewise linear; either ReLU or Max-pooling is a suitable choice. We denote the output matrix of the $\gamma$-stack as $\Phi$ and the output vector of the $\beta$-stack as $\Psi$, respectively. To model the outer maximization in (6), the output of the model is the maximum element of the dot product between $\Phi$ and $\Psi$. Finally, the $\beta$-stack is constrained so that the first layer has non-positive weights and each subsequent layer has non-negative weights; this enforces the desired properties of a GVP listed in Theorem 3, such as convexity on $\beta$ [3]. In general, the $\gamma$-stack represents the functions $\{F_p\}_{p=1}^{Q}$ and the $\beta$-stack models the function $G$ from Theorem 2.

We can summarize the properties of DSM as follows. Let $a^i$ the $i$-th column of the matrix $A$. Then we may define $\mathcal{H}(A) := \{\eta_A(\cdot) \mid \eta_A \text{ is a DSM and } \eta_A(\gamma, a^i) \leq \gamma_i \ \forall \gamma \in \mathcal{C}, i \in [\![n]\!]\}$ to be the class of functions that can be represented by a DSM, subject to what we might call dual feasibility constraints on their outputs.

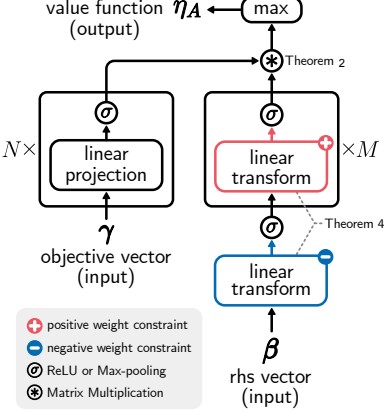

Figure 1: Dual-Stack Model (DSM)

**Theorem 3.** *Any function $\eta_A \in \mathcal{H}(A)$ has the following properties:*

1. *$\eta_A(\gamma, \cdot)$ is piecewise linear, convex, and monotonically decreasing for every fixed $\gamma \in \mathcal{C}$.*

2. *$\eta_A(\cdot, \beta)$ is piecewise linear for every fixed $\beta \in \mathcal{B}$.*

3. *$\eta_A(\gamma, \beta) \leq h_A(\gamma, \beta)$ for every fixed $\beta \in \mathcal{B}$ and $\gamma \in \mathcal{C}$.*

This result shows that, for fixed inputs, $h_A$ is upper-bounded by the true GVF. In fact, we can show something even stronger.

**Theorem 4.** *For any fixed $A \in \mathbb{R}^{m \times n}$, $h_A \in \mathcal{H}(A)$, and moreover $h_A$ is pointwise larger than all other elements of $\mathcal{H}(A)$.*

One way to interpret Theorem 4 is that there exists some DSM architecture whereby we can recover the GVF by setting the weights in such a way that we recover the pointwise maximum across infinitely many points in $\mathcal{B} \times \mathcal{C}$. We can now sharpen this result to show that it suffices to restrict attention to some finite subset of these points. For any given $\mathcal{M} \in \mathbb{Z}_{\geq 0}^{M}$ and $\mathcal{N} \in \mathbb{Z}_{\geq 0}^{N}$, define $\text{DSM}(\mathcal{M}, \mathcal{N})$ to be the class of functions represented by a DSM whose $\gamma$-stack and $\beta$-stack have layers with $\mathcal{M}$ and $\mathcal{N}$ neurons each, respectively.

**Theorem 5.** *There exists some $\mathcal{M} \in \mathbb{Z}_{+}^{M}$, some $\mathcal{N} \in \mathbb{Z}_{+}^{N}$, a finite set $\bar{\mathcal{C}} \subsetneq \mathcal{C}$, a finite set $\bar{\mathcal{B}} \subsetneq \mathcal{B}$, and some $\eta' \in \text{DSM}(\mathcal{M}, \mathcal{N})$ such that $\eta'(\gamma, \beta) = h_A(\gamma, \beta)$ for all $\beta \in \bar{\mathcal{B}}, \gamma \in \bar{\mathcal{C}}$. Moreover, this same $\eta'$ necessarily satisfies $\eta'(\gamma, \beta) = h_A(\gamma, \beta)$ for all $\beta \in \mathcal{B}$ and $\gamma \in \mathcal{C}$.*

Finally, we reframe this existential result as an optimization problem; this will form the basis for the unsupervised training framework we develop in Section 5.

**Corollary 6.** *Take the $\mathcal{M}$, $\mathcal{N}$, $\bar{\mathcal{B}}$, and $\bar{\mathcal{C}}$ that Theorem 5 guarantees must exist. Denote the parameters of a DSM model with $\theta$. Then, $h_A$ is the unique solution of*

$$\max_{\eta^\theta \in \text{DSM}(\mathcal{M}, \mathcal{N})} \sum_{\gamma \in \bar{\mathcal{C}}} \sum_{\beta \in \bar{\mathcal{B}}} \eta^\theta(\gamma, \beta) \tag{7a}$$

$$s.t. \ \eta^\theta(\gamma, a^i) \leq \gamma_i \quad \forall \gamma \in \bar{\mathcal{C}}, i \in [\![n]\!]. \tag{7b}$$

We refer the reader to Figure 2 for an illustration of how, taken together, the results of this section permit us to learn a good approximation of the GVP. In addition, we highlight that typical "universal approximation theorems" [35] apply *over bounded input domains*, whereas here $\mathcal{C}$ and $\mathcal{B}$ may be unbounded. However, the above approach shows that we can attain $h_A$ without this assumption.

## 5  Learning Generalized Linear Programming Value Functions

For context, we begin by describing a standard supervised training method to approximate the function $h_A$ using a neural network with parameters $\theta$. First, we generate some training data set

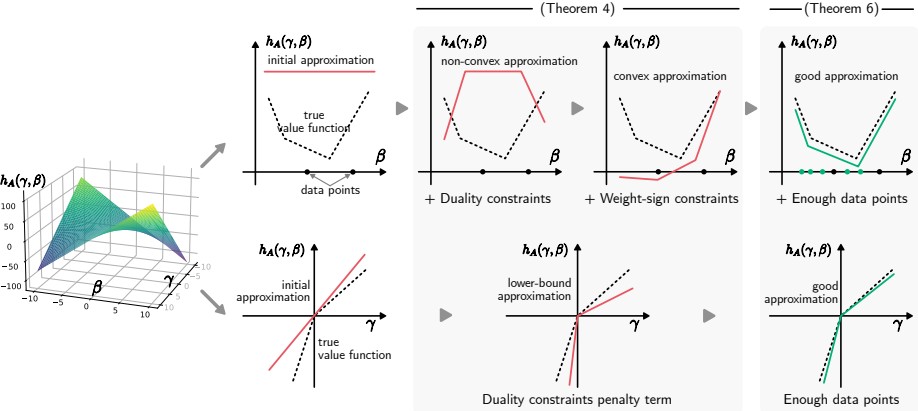

Figure 2: An illustration of how we learn a GVF, looking at slices along the constraint bounds (Top) and objective coefficients (Bottom). Given only data points $\bar{\mathcal{B}} \times \bar{\mathcal{C}}$ and no constraints, either (7b) or weight-sign, maximizing $\sum_{\gamma \in \bar{\mathcal{C}}} \sum_{\beta \in \bar{\mathcal{B}}} \eta^\theta(\gamma, \beta)$ will yield a poor initial approximation. Adding constraints (7b) will force the function to be smaller than $h_A$ at certain "anchor" points which are distinct from the input data points, at which the function will still tend to be large. Adding the weight-sign constraints will force the function to be convex in terms of the constraint bounds, and will therefore produce an approximation that lower bounds $h_A$. Theorem 5 then tells us that, with sufficiently many data points to start, we will eventually recover $h_A$ directly.

$\mathcal{D} := \{(\gamma^i, \beta^i; h_A(\gamma^i, \beta^i))\}_{i=1}^{|\mathcal{D}|}$. Then, we optimize our parameters $\theta$ by minimizing a loss function (such as an $\ell_2$-distance) between the output of the NN with our data. Notably, this is an unconstrained optimization problem. However, data generation may be expensive: in general, we must solve an linear programming problem, $LP(\gamma^i, \beta^i)$, for every training data point $i \in [\![|\mathcal{D}|]\!]$.

## 5.1 An Unsupervised Learning Approach

Corollary 6 tells that we do not actually need labeled data to learn the GVF. Of course, there is a trade-off here: directly applying the result requires us to somehow compute the sets $\bar{\mathcal{B}}$ and $\bar{\mathcal{C}}$, and also requires solving a constrained optimization problem in (7) (as opposed to a typical unconstrained learning problem). To address the first issue, we propose using two subsets $\mathcal{D}_b \subsetneq \mathcal{B}$ and $\mathcal{D}_c \subsetneq \mathcal{C}$ to use in lieu of $\bar{\mathcal{B}}$ and $\bar{\mathcal{C}}$ respectively, which may come from training data or be randomly generated; this is further detailed for a specific application in Section 7. To address the second issue, we introduce a penalty term to the objective to model the constraints (7b) in a "soft" manner. This leaves us with the following unconstrained, unsupervised learning problem:

$$\min_{\eta^\theta \in \text{DSM}(\mathcal{M}, \mathcal{N})} \sum_{\gamma \in \mathcal{D}_c} \sum_{\beta \in \mathcal{D}_b} -\eta^\theta(\gamma, \beta) + \mu \sum_{\gamma \in \mathcal{D}_c} \sum_{i \in [\![n]\!]} \max\{\eta^\theta(\gamma, a^i) - \gamma_i, 0\}, \qquad (8)$$

where $\mu \in \mathbb{R}_+$ is the penalty coefficient. We can motivate both of the alterations introduced above with the following corollary to Theorem 5. To enforce the constraint $\eta^\theta \in \text{DSM}(\mathcal{M}, \mathcal{N})$, we take the positive (respectively, negative) absolute value of the weights for a nonnegative (resp. nonpositive) weight-sign constraint.

**Corollary 7.** *Given any nonempty subsets $\mathcal{D}_b \subseteq \mathcal{B}$ and $\mathcal{D}_c \subseteq \mathcal{C}$, there exists a sufficient large $\mu$ such that $h_A$ is an optimal solution of (8).*

Note that, while (7) has $h_A$ as its unique optimal solution, in general we cannot guarantee that (8) has an unique optimal solution, meaning that we may recover an optimal solution that deviates from $h_A$ on points outside the training set. However, Corollary 7 ensures that any solution of (8) is at least as good as $h_A$ at $(\gamma, \beta) \in \mathcal{D}_c \times \mathcal{D}_b$[1]. Moreover, in Section 5.2 we will provide computational evidence that a near-optimal solution of (8) is empirically a good approximation of $h_A$. Corollary 7 also justifies why we elect to use an $\ell^1$ penalty term in (8), rather than a smooth penalty or Lagrangian

---

[1]In fact, because of Theorem 3, any solution of (8) is at least as good as $h_A$ at every point of $\{t(\gamma, \beta) | t \geq 0, (\gamma, \beta) \in \mathcal{D}_c \times \mathcal{D}_b\}$.

multipliers. If we were to modify (8) to use an $\ell^2$ penalty term, for example, an analogous version of Corollary 7 need not hold: roughly speaking, $h_A$ is the function that satisfies most of (7b) at equality, whereas an $\ell^2$ penalty term would tend to push optimal solutions away from the boundary of the feasible region of (7). We refer the interested reader to [40, Chapter 17] for a detailed discussion of exact-inexact or smooth-nonsmooth penalty terms.

## 5.2 Penalty Coefficient Update Strategies

Now, we use standard techniques from nonlinear optimization [40, Chapter 17] to develop a heuristic scheme for updating our penalty term $\mu$, depicted in Algorithm 1. In practice, the optimization problem (7) might be numerically unstable. For example, if we initialize $\mu_0$ to a small value, then the first term $\sum_{\gamma \in \mathcal{D}_c, \beta \in \mathcal{D}_b} \eta_A^\theta(\gamma, \beta)$ is weighed far more than the penalty term, and so the optimal solution will tend towards $-\infty$. We resolve this by using upper bounds, $u$, on $h_A$ as "tether" points. For each training data point, we choose a simple upper bound of the optimal objective (see Section 7 and Appendix E for details).

We parameterize our scheme on the penalty update strategy, which is perhaps the most important factor. A common choice would be the linear update strategy, which means $\mu$ is scaled up by a constant factor $\nu$ at each step, i.e., $\texttt{update}(\mu) = \nu\mu$. However, we propose a more computationally effective "adaptive update" strategy where we update $\mu$ based on how many constraints (7b) the current solution $\eta^t$ satisfies. In particular, $\texttt{update}(\mu) = (2 - \texttt{percentage\_cons\_satisfied}) \cdot \mu$.

---

**Algorithm 1** Learning GVF with objective upper bounds

1: **procedure** LEARNINGGENERALIZEDLPVF($A, \mathcal{D}_c, \mathcal{D}_b, u, T, \mu_0, \texttt{update}$)
2:    Initialize $\eta^0$
3:    **for** $t \leftarrow 0, \dots, T-1$ **do**
4:       $\eta^{t+1} \leftarrow \text{argmin} \sum_{\gamma \in \mathcal{D}_c} \sum_{\beta \in \mathcal{D}_b} (u_{\gamma,\beta} - \eta^\theta(\gamma, \beta))^2 + \sum_{\gamma \in \mathcal{D}_c} \sum_{i \in [\![n]\!]} \mu_t \cdot \max\{\eta^\theta(\gamma, a^i) - \gamma_i, 0\}$
5:       $\mu_{t+1} \leftarrow \texttt{update}(\mu_t)$
6:    **Return** $\eta^T$.

---

Note that, since the minimization problem in line 4 of Algorithm 1 is non-convex, we cannot guarantee a globally optimal solution $\eta^{t+1}$. Therefore, in practice, we solve the minimization subproblem until some criterion is met, e.g., the gradient is sufficiently small or we reach a prescribed iteration limit.

## 5.3 Guaranteeing an Under-Approximation

By Theorem 3, any feasible solution for (7) lower bounds the GVF we are attempting to learn. However, our methods laid out in this section do not guarantee this property for two reasons. First, the penalty method treats the constraints (7b) as soft constraints rather than hard constraints, and thus they may not be fully satisfied. Second, we are approximating the set $\bar{\mathcal{C}}$ from Corollary 6 with a training set $\mathcal{D}_c$, and therefore our solution may not provide a lower bound at other objective vectors $\gamma$. To resolve this, we can scale the function down as much as needed to guarantee the constraints (7b), with the expectation that our learning method produces a solution that is not too far off from being dual feasible. However, we can do better if we only need to provide a valid lower bound for a single objective $\gamma$, which is often the case (see Section 6 below). In this way, we can train a single DSM approximation and reuse it across many objectives by suitably postprocessing it for each. We describe this procedure in Algorithm 2.

---

**Algorithm 2** Post-processing to guarantee the lower-bounding property

1: **procedure** POST-PROCESSING($c, \Psi^A, \phi^A$)
2:    $\Phi^c \leftarrow \phi^A(c) \in \mathbb{R}^{p \times N}$
3:    **for** $j \leftarrow 1, \dots, p$ **do**
4:       $\Phi_j^c \leftarrow \Phi_j^c \cdot \left( 1/\max_i \left\{ \frac{c_i}{(\Phi_j^c)^T \Psi_i^A} \; \middle| \; (\Phi_j^c)^T \Psi_i^A \geq c_i \right\} \right)$
      **return** $\Phi^c$

---

# 6  A GVF-Based Heuristic for Two-Stage MILPs

As an application of the learning method we developed in Section 5, we propose a heuristic method for two-stage MILPs with continuous second-stage variables (4). The main idea is to replace each of the second-stage subproblems with the corresponding LPVFs from our learned function: that is, we use our approximation to represent $h_A(d^k, b^k - T^k x^1)$ in (5) for each $k$. This yields a fast heuristic for two reasons. First, our learned approximation is piecewise linear convex when restricted to a fixed objective, meaning that it can be efficiently modeled inside a larger optimization problem as an LP [3]. Second, the number of variables of this LP scales with the number of neurons in the DSM, typically much smaller than a second-stage subproblem LP. In practice, this enables the heuristic to run faster than solving the LP relaxation of (5), despite maintaining integrality of the first-stage variables.

Given a learned DSM representing the function $\eta_A^\theta$, denote by $W_\Psi^0, W_\Psi^1, \ldots, W_\Psi^M$ the weights of the constraint bounds stack, where $W_\Psi^0 \leq 0$ and $W_\Psi^1, \ldots, W_\Psi^M \geq 0$. For a fixed objective coefficient $c \in \mathcal{C}$ and the output of the objective-stack $\Phi^c$ (post-processed as in Section 5.3), we can model the set $\{\zeta \mid \zeta \geq \eta_A^\theta(c, \beta)\}$ as

$$\text{DSM}_\theta(c, \beta) = \left\{ \zeta \ \middle| \ \exists z \text{ s.t. } z_1 \geq \sigma(W_\Psi^0 \beta), \ z_{i+1} \geq \sigma(W_\Psi^i z_i) \ \forall i \in [\![M]\!], \ \zeta \geq \Phi_i^c \cdot z_M \ \forall i \in [\![p]\!] \right\}.$$

Our heuristic is then to solve the following problem to obtain a solution for the first-stage variables:

$$\min_{x^1, \zeta^1, \ldots, \zeta^K} \left\{ c \cdot x^1 + \sum_{k \in [\![K]\!]} \zeta^k \ \middle| \ \zeta^k \in \text{DSM}_\theta(d^k, b^k - T^k x^1) \ \forall k \in [\![K]\!], x \in X \right\} \quad (9)$$

Note that (9) is simply (5) with the LPVFs replaced by our learned model. Since $\eta_A^\theta(c, \beta) \leq h_A(c, \beta)$ for all $c \in \mathcal{C}, \beta \in \mathcal{C}$, the objective value of (9) is a dual bound for the original problem. Once we compute optimal values of the first-stage variables $x^*$, we can then recover the second-stage variable values by solving each of the second-stage LP subproblems independently with fixed $x^*$, yielding the full solution. Algorithm 3 describes the full method.

---

**Algorithm 3** GVF-based heuristic for two-stage MILPs with continuous second-stage variables

---

1: **procedure** GVFBASEDHEURISTIC($A, c, \{T^k\}_{k \in [\![K]\!]}, \{b^k\}_{k \in [\![K]\!]}, \{d^k\}_{k \in [\![K]\!]}, \Psi, \phi$)
2:     **for** $k \leftarrow 1, \ldots, K$ **do** $\Phi^{d^k} \leftarrow$ POST-PROCESSING($c, \Psi, \phi$)
3:     $((x^*)^1, \{\zeta^{*k}\}_{k \in [\![K]\!]}) \leftarrow$ optimal solution of (9) with $W_\Psi, \{\Phi^{d^k}\}_{k \in [\![K]\!]}$
4:     **for** $k \leftarrow 1, \ldots, K$ **do**
5:         $(x^*)^{2,k} \leftarrow \arg\min_{x^{2,k}} \{d^k \cdot x^{2,k} \mid A x^{2,k} \leq b^k - T^k x^{*1}, x^{2,k} \geq 0\}$
        **return** $x^*$

---

# 7  Computational Results

In this section, we computationally evaluate[2] both the approximation quality of the learning method described in Section 5 and the effectiveness of the heuristic for two-stage problems from Section 6.

We evaluate these methods on the *uncapacitated facility location* (UFL) [54]. This is a deterministic two-stage problem, in which we first select $n_f$ facilities to open, and allocate each of $n_c$ customers to an open facilities. We consider two classes of instances, *Euclidean* and *KG*, both with $n_c = n_f$. In both cases, we take a set of objective and right-hand side vectors from one instance for training and a second, different, set from five instances for testing (more details are provided in Appendix C).

To produce the training data, we take all or some customer allocation costs from the UFL training instances as our objective coefficient dataset $\mathcal{D}_c$. Then, for each such cost vector, we generate $\lfloor n_f/10 \rfloor$ points uniformly at random between $[0, 1]$ for our constraint bound dataset $\mathcal{D}_b$. Thus, the total size of the training data is $\lfloor |\mathcal{D}_c| \cdot n_f/10 \rfloor$. If all customer costs are selected for $\mathcal{D}_c$, this is

---

[2]All code for the experiments can be found at https://github.com/google-research/google-research/tree/master/learning_gvf.

$\lfloor |\mathcal{I}| \cdot n_c \cdot n_f / 10 \rfloor$ where $\mathcal{I}$ is the set of training instances. Note that we do not use the facility costs at all for training. To improve training stability, we normalize the customer allocation costs by their mean in the training data. For the training dataset, we choose an upper bound of the GVF to be 2, which is an upper bound for the largest possible assignment cost in the training dataset after normalization.

To learn each GVF, we run a total of $T = 40$ iterations in Algorithm 1, at each iteration, we solve (8) by performing 100 steps of the Adam algorithm [31]. For DSM, we select the model within the $T$ iterations that satisfies at least $98\%$ of the constraints (7b) from the training dataset with lowest training objective function (8). Details of hyperparameter tuning for DSMs and DenseNets are provided in Appendix E. In addition to the numerical study for UFL, we include experiments on the Stochastic Capacitated Facility Location (SCFL) in Appendix D.

## 7.1 Learning Method

Arguably, we would expect that the lack of supervised data would make the Dual-Stack Model more difficult to train than a standard supervised learning approach, especially considering that it also has training constraints and convexity requirements. On the other hand, our theoretical results suggest that these same requirements can help the model take the general shape of a GVF. Indeed, we observe in Table 1 that our approach produces a model that is comparable or better than a standard ReLU network (see Appendix E for DenseNet) and Random Forest Regressors in terms of how well it approximates the GVF. Unlike with DSM, these baselines require solving an LP for each training point to produce labels. The number of LPs can grow large, though they can be solved in an embarrassingly parallel manner, and in the case of UFL we can use an efficient greedy algorithm given that the LP reduces to fractional knapsack. We report an *a posteriori* metric, the True Relative Error, defined as the gap between the model and the GVF, i.e., $|\eta(\gamma, \beta) - h_A(\gamma, \beta)| / \max\{\eta(\gamma, \beta), h_A(\gamma, \beta)\}$, averaged across all $\gamma, \beta$ in the test set. The Lower Bound in Table 1 shows the percentage of constraints (7b) satisfied for either the training or test set.

We see that, on all but one instance family, we are able to train a DSM that attains a lower True Relative Error than both DenseNet and Random Forest. Moreover, the training times between DSM and DenseNet are roughly comparable. We also highlight that the Euclidean instances are relatively harder to learn than the KG instances; this is observable for all models.

Table 1: Comparison between Dual-Stack Model and DenseNet in Learning GVF.

| Class of GVF | | Dual-Stack Model | | | | DenseNet | | | Random Forest | |
| | Train Time (s) | True Rel. Error | Train Lower Bound | Test Lower Bound | Data Label Time (s) | Train Time (s) | True Rel. Error | Train Time (s) | True Rel. Error |
|---|---|---|---|---|---|---|---|---|---|---|
| KG  250 | 157.20 | 1.52 % | 98.05 % | 25.71 % | 5.41 | 164.31 | **1.09 %** | 73.4 | 6.05 % |
| KG  500 | 1288.44 | **1.14 %** | 98.76 % | 24.70 % | 20.61 | 1439.27 | 1.64 % | 1092.50 | 6.18 % |
| KG  750 | 1082.48 | **0.87 %** | 98.73 % | 33.19 % | 13.27 | 1125.98 | 4.64 % | 1751.73 | 6.25 % |
| Euclidean  100 | 7.57 | **33.88 %** | 98.07 % | 58.90 % | 1.17 | 7.74 | 39.29 % | 37.93 | 41.41 % |
| Euclidean  200 | 62.96 | **33.78 %** | 98.01 % | 67.45 % | 4.98 | 69.66 | 43.12 % | 77.48 | 37.22 % |
| Euclidean  300 | 161.25 | **31.89 %** | 99.41 % | 76.50 % | 8.26 | 201.31 | 33.25 % | 375.60 | 39.26 % |

Table 1 examines the capacity of the DSM model to learn GVFs. To illustrate the scalability of DSMs, we train the model on LPs of various sizes. We use the allocation cost vectors of all customers for $\mathcal{D}_c$ in all cases, except for KG 750 where we select 200 customers for $\mathcal{D}_c$ due to memory limitations. Although training time naturally increases with instance size, both the True Relative Error and Lower Bounds remain stable for KG instances and exhibit only a slight increase in the Euclidean cases. This observation supports the scalability of the DSM approach.

## 7.2 Heuristic for Two-Stage Problems

We evaluate our heuristic on our generated UFL instances by comparing its performance against that of a state-of-the-art open-source MILP solver, SCIP [20]. We compare with the best feasible solution found by the solver within the time limit specified in the column "MIP Solve Time (s)" in Table 2 for KG, or the optimal solution for Euclidean instances. For KG, even solving the LP relaxation for the full model can take a few minutes. On the other hand, despite being an MILP, solving the model (9) is very fast, often taking less than a second, particularly because we only require binary constraints on the first-stage variables. After solving this MILP, the second-stage solutions are recovered by simply taking the closest open facility. Denoting the objective value of the heuristic feasible solution by $v^*$ and the optimal value of (9) by $V$—which is a dual bound to the original problem due to our under-approximation guarantee—the Provable Gap is computed as $(v^* - V)/v^*$, which is an upper

bound of the true gap. The Gap to MILP is computed as $(v^* - \bar{v})/v^*$, where $\bar{v}$ is the objective value of the MILP baseline. A negative value of Gap to MILP means that the solution returned by our heuristic is better than the solution returned by SCIP within the time limit. We solve 5 instances from each class and report the mean and standard deviation for each metric across these selected instances in each class. When reporting time, we use '>' to signal the solver reaches a time limit. We also compare our method with a heuristic based on Benders Decomposition, which mimics the DSM heuristic except that we use Benders cuts instead of the GVF. For each instance, we first iteratively generate a number of Benders cuts for all subproblems, and then solve one MILP with the inclusion of all generated optimality cuts. For each UFL instance, we set the maximum number of Benders LP iterations to match the total number of facilities, while the time limit for solving the MILPs with cuts is fixed at 1 minute. In the Euclidean case, the Benders heuristic was able to produce an optimal solution for all instances tested. We observe from Table 2 that our approach produces significantly better solutions than our full model baseline for large KG instances, though not for Euclidean ones. This may be because the Euclidean instances are more difficult to learn as observed in Table 1, both for DSM and the dense model. We also note that the Euclidean instances are much *easier* to solve to optimality than the KG instances, meaning that a fast heuristic for them is of relatively lesser value.

Table 2: DSM Heuristic Solver on UFL Instances.

| Instances | | DSM Heuristic Solver | | | Full Model Solver | | Benders Heuristic |
|---|---|---|---|---|---|---|---|
| | | Solve Time (s) | Provable Gap (%) | Gap to MILP (%) | Gap to Benders | LP Relaxation Solve Time (s) | MILP Solve Time (s) | Solve Time (s) |
| **KG** | 250 | $0.070 \pm 0.001$ | $< 27.24 \pm 0.76$ | $12.55 \pm 0.80$ | $11.54 \pm 0.01$ | $3.42 \pm 0.31$ | $> 30$ | $63.3 \pm 0.2$ |
| | 500 | $0.111 \pm 0.008$ | $< 14.53 \pm 0.84$ | $-3.79 \pm 1.05$ | $13.14 \pm 0.01$ | $36.07 \pm 1.52$ | $> 60$ | $92.1 \pm 0.5$ |
| | 750 | $0.313 \pm 0.004$ | $< 17.66 \pm 0.40$ | $-55.44 \pm 1.34$ | $14.25 \pm 0.01$ | $> 90$ | $> 90$ | $189 \pm 3.6$ |
| **Euclidean** | 100 | $0.018 \pm 0.001$ | $< 74.40 \pm 3.70$ | $23.32 \pm 6.25$ | $23.32 \pm 6.25$ | $0.57 \pm 0.02$ | $0.20 \pm 0.02$ | $0.031 \pm 0.001$ |
| | 200 | $0.032 \pm 0.001$ | $< 94.52 \pm 0.89$ | $43.59 \pm 9.10$ | $43.59 \pm 9.10$ | $0.60 \pm 0.04$ | $0.67 \pm 0.04$ | $0.083 \pm 0.016$ |
| | 300 | $0.048 \pm 0.002$ | $< 96.22 \pm 0.56$ | $47.98 \pm 8.60$ | $47.98 \pm 8.60$ | $1.41 \pm 0.04$ | $1.55 \pm 0.04$ | $0.174 \pm 0.019$ |

# 8 Conclusion

JIn this study, we provide a structural characterization of the GVF, inspiring an NN architecture and unsupervised method that approximate it well. Additionally, we utilize this framework to develop a fast heuristic method for two-stage MILPs with continuous second-stage variables, effective for some large-scale instances. We will conclude this paper by highlighting two areas for future work, where we believe that the techniques presented in this paper could be sharpened or otherwise improved.

First, we believe that our training objective could be further improved. During training, we must balance two terms in our loss function: one that rewards fitting the data well, and another one that (softly) constrains the function to be below the true value function. Finding a stable balance between these two terms appears to be one of the most challenging parts of training. We overcome this limitation by proposing an adaptive update method for the penalty, tuning the initial hyperparameter well, and proposing a good stopping criterion. However, even with all these measures, we still observe occasional instability and believe that there is room for improvement, such as developing an adaptive stopping rule.

A second direction for future work is to improve the generalization of our method to objectives not seen in the training set. While our theoretical results guarantee that the constraints in (7) are sufficient to produce a function that lower bounds the GVF, this may require an exponential number of constraints. In practice, we only enforce this constraint for those objective vectors appearing in the training data, which does give us a lower bounding guarantee for arbitrary right-hand sides and objective vectors from outside the training set. We observe empirically that with sufficient samples, our method performs well on out-of-sample objectives, but we do note that our Test Lower Bound column in Table 1 differs from the Train Lower Bound column by a significant amount.

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

# A Complete Proof for GVF Representation Theorem

In this section, we provide the complete proof for the GVF Representation Theorem (Theorem 2). The following lemmas are inspired by [11, Lemma 6.1, Theorem 6.2 and Theorem 6.3]. These results from Blair and Jeroslow are originally stated for the value functions of integer programs. We borrow their techniques and extend these results for the GVFs. In the following proofs, we use English letters, e.g., $b$ or $c$, to indicate a fixed vector, while Greek letters, e.g., $\beta$ or $\gamma$, are used to signal inputs of some functions and can vary.

## A.1 Lemma 8

Lemma 8 states that there is a *finite* set $D$ of *improving directions*: given a suboptimal feasible point $x \in X(\beta)$, we can improve $x$ by moving along some $r \in D$. Notably, the set $D$ depends only on $A$ and not on $\gamma$ or $\beta$, thus setting a basis for constructing the function $h_A$.

**Lemma 8.** *There exists a finite set $D \subset \mathbb{R}^n$ such that, for any $\beta \in \mathcal{B}$, $\gamma \in \mathcal{C}$, and $x \in X(\beta)$, exactly one of the following is true:*

1. $\gamma \cdot x = h_A(\gamma, \beta)$, *or*

2. $\exists \epsilon > 0$ *and* $r \in D$ *such that* $x + \epsilon r \in X(\beta)$ *and* $\gamma \cdot r < 0$.

*Proof.* Let $\{B_1, B_2, \ldots, B_k\} \subset \mathbb{R}^{(n-1) \times n}$ be the set of all full rank submatrix of $[A^T \ I_n]^T$. Since $B_i$ is full rank, there exists $r_i \in \mathbb{R}^n$ and $\|r_i\|_2 = 1$ such that for every $t \in \mathbb{R}$, $tr_i$ is a solution of the system $B_i r = 0$. Let $D := \{\pm r_i | \forall i \in [\![k]\!]\}$. We show that the finite set $D$ satisfies the condition of Lemma 8.

We first show that, for any $\beta \in \mathcal{B}$, every extreme ray of $X(\beta)$ (normalized so that its $\ell^2$-norm is one) belongs to $D$. Let $v$ be an extreme ray of $X(\beta)$. Then, by definition, $v$ must satisfy at equality exactly $n-1$ inequalities from $Av \leq 0$ and $v \geq 0$. Therefore, there exists a matrix $B_i$ such that $B_i v = 0$ and thus $v \in D$.

Consider $x \in X(\beta)$ a feasible solution of $LP(\gamma, \beta)$. Let $P$ and $V$ denote the set of extreme points and extreme rays of $X(\beta)$ respectively. By Carathéodory's Theorem [12], for any feasible point $x \in X(\beta)$, there exists $\{\lambda_p \geq 0\}_{p \in P}$ where $\sum_{p \in P} \lambda_p = 1$ and $\{\rho_v \geq 0\}_{v \in V}$ such that $x = \sum_{p \in P} \lambda_p p + \sum_{v \in V} \rho_v v$. If there exists $v' \in V$ such that $\rho_{v'} > 0$ and $\gamma \cdot \rho_{v'} > 0$, then $x$ cannot be a optimal solution of $LP(\gamma, \beta)$ because $x - \rho_{v'} v'$ has a smaller objective and $-v' \in D$ satisfies the second condition of Lemma 8. Without loss of generality, from now on, we assume that $x = \sum_{p \in P} \lambda_p p$.

Since $x$ is a feasible solution, $x$ is either a optimal solution of $LP(\gamma, \beta)$ or not. If $x$ is the optimal solution, then $\gamma \cdot x = h(\gamma, \beta)$. Otherwise, by definition of $\mathcal{B}$ and $\mathcal{C}$, $LP(\gamma, \beta)$ has an basic optimal solution $p^* \neq x$ such that $\gamma \cdot p^* < \gamma \cdot x$. Since $p^* - x = \sum_{p \neq p^*} \lambda_p p^* + (1 - \lambda_{p^*}) p^*$, we have $p^* - x \in \text{span}(D)$. By our choice of $D$, if $r \in D$ then $-r \in D$, thus there exists $r \in D$ such that $\gamma \cdot r < 0$ and $\exists \epsilon > 0$ and $r \in D$ such that $x + \epsilon r \in X(\beta)$. $\square$

## A.2 Lemma 9

Using the improving directions from Lemma 8, Lemma 9 produces a finite set of extreme rays that generate the objective coefficient space $\mathcal{C}$. By this lemma, the value of $h_A(\gamma, \beta)$ can always be written as a conic combination of $\{h_A(c_q, \beta)\}_{q \in [\![Q]\!]}$.

**Lemma 9.** *There exists a finite set $C := \{c_1, c_2, \ldots, c_Q\} \subset \mathcal{C}$ such that for any $c \in \mathcal{C}$, $b \in \mathcal{B}$,*

$$\begin{pmatrix} \gamma \\ h_A(c, b) \end{pmatrix} \in cone \left( \begin{pmatrix} c_1 \\ h_A(c_1, b) \end{pmatrix}, \ldots, \begin{pmatrix} c_Q \\ h_A(c_Q, b) \end{pmatrix} \right) \tag{10}$$

*Proof.* Let $D$ be the finite set satisfying the condition in Lemma 8. For every subset $R \subset D$, we define

$$C(R) = \{c \in \mathcal{C} | c \cdot r \geq 0 \ \forall r \in R\}.$$

Since $C(R)$ is a polyhedral cone for every $R \subset D$, we denote $E(R)$ to be the finite set of extreme rays of $C(R)$. We show that $C := \cup_{R \subset D} E(R)$ and denote its elements as $\{c_1, c_2, \ldots, c_Q\}$ satisfies the condition stated in Lemma 9.

For any $\gamma \in \mathcal{C}$ and $\beta \in \mathcal{B}$, since $LP(\gamma, \beta)$ has a finite optimal solution, there must exists $x^*$ such that $h(\gamma, \beta) = \gamma \cdot x^*$. Let $R^* := \{r \in D \mid \exists \epsilon > 0,\ x^* + \epsilon r \in X(\beta)\}$ be the set of all feasible improving directions at $x^*$.

**Claim 1:** *By our choice of $R^*$, we must have $\gamma \in C(R^*)$.* By contradiction, suppose that $\gamma \notin C(R^*)$, then there must exist $r^* \in R^*$ such that $\gamma \cdot r^* < 0$. However, by definition of $R^*$, there exists $\epsilon > 0$ such that $x^* + \epsilon r^*$ is a feasible solution of $LP(\gamma, \beta)$. In addition, we have $\gamma \cdot (x^* + \epsilon r^*) = \gamma \cdot x^* + \epsilon \gamma \cdot r^* < \gamma \cdot x^*$, which contradicts the fact that $x^*$ is an optimal solution of $LP(\gamma, \beta)$. Hence, we derive that $\gamma \in C(R^*)$. Moreover, since $C(R^*)$ is a polyhedral cone whose extreme rays $E(R^*)$ belongs to $C$ by our construction of $C$, we have there exists $\alpha \geq 0$ such that $\sum_{q \in \llbracket Q \rrbracket} \alpha_q c_q = \gamma$.

**Claim 2:** *For every $c \in C(R^*)$, we have $h(c, \beta) = c \cdot x^*$.* Since $x^* \in X(\beta)$, we have $h(c, \beta) \leq c \cdot x^*$. Moreover, because $x^*$ is optimal solution of $LP(\gamma, \beta)$, we also have $\gamma \cdot r \geq 0$ for every $r \in R^*$. By contradiction, suppose there exists $c \in C(R^*)$ such that $h(c, \beta) < c \cdot x^*$. By Lemma 8 and $x^*$ is not an optimal solution of $LP(\gamma, \beta)$, there exists $\epsilon > 0$ and $r \in D$ such that $x^* + \epsilon r \in X(\beta)$ and $c \cdot r < 0$. However, since $x^* + \epsilon r \in X(\beta)$, we must have $r \in R^*$, and by definition of $C(R^*)$, every $c \in C(R^*)$ must satisfy $c \cdot r \geq 0$, which contradicts with our conclusion that $c \cdot r < 0$. Therefore, $h(c, \beta) = c \cdot x^*$ for every $c \in C(R^*)$.

Finally, we have that

$$\sum_{q \in \llbracket Q \rrbracket} \alpha_q h(c_q, \beta) = \sum_{q \in \llbracket Q \rrbracket} \alpha_q c_q \cdot x^* = \left( \sum_{q \in \llbracket Q \rrbracket} \alpha_i c_q \right) \cdot x^* = \gamma \cdot x^* = h(\gamma, \beta).$$

This is because from Claim 1, $\gamma \in C(R^*)$, without loss of generality, we assume that $\alpha_q = 0$ for every $c_q \notin E(R^*)$. Therefore, we either have $\alpha_q = 0$ or $h(c_q, \beta) = c_q \cdot x^*$. Thus, $\sum_{q \in \llbracket Q \rrbracket} \alpha_i h(c_q, \beta) = \sum_{q \in \llbracket Q \rrbracket} \alpha_i c_q \cdot x^*$. $\qquad \square$

### A.3 Lemma 10

For ease of notation, we denote $g_q(\beta) := h_A(c_q, \beta)$ for every $q \in \llbracket Q \rrbracket$. A function $g_q$ is obtained from $h_A$ by fixing the objective coefficients to $c_q$. Therefore, $g_q(\beta)$ is an LPVF (parameterized by constraint bounds), and thus a convex piecewise linear function.

While Lemma 9 shows that $h_A$ can be represented by a finite set of LPVFs $\{g_q\}_{q \in \llbracket Q \rrbracket}$, the linear combination coefficients of such representation depend on $b \in \mathcal{B}$ and $c \in \mathcal{C}$ and are unclear how to obtain. The following Lemma 10 gives a more explicit characterization of the linear combination coefficients.

**Lemma 10.** *For every $\gamma \in \mathcal{C}$ and $\beta \in \mathcal{B}$, we have*

$$h_A(\gamma, \beta) = \max_\alpha \sum_{q \in \llbracket Q \rrbracket} \alpha_q g_q(\beta)$$

$$s.t \sum_{q \in \llbracket Q \rrbracket} \alpha_q c_q = \gamma \qquad\qquad (LP_\alpha(\gamma, \beta))$$

$$\alpha \geq 0.$$

*Proof.* Let $x^*$ be the optimal solution of $LP(\gamma, \beta)$ and $x^i$ be the optimal solution of $LP(c_i, \beta)$ for every $i \in \llbracket N \rrbracket$. Since $x^* \in X(\beta)$ is always a feasible solution of $LP(c_q, \beta)$, we have

$$c_i \cdot x^* \geq c_i \cdot x^i \ \forall q \in \llbracket Q \rrbracket$$

Therefore, for every $\alpha$ such that $\sum_{q \in \llbracket Q \rrbracket} \alpha_q c_q = \gamma$, we have

$$h(\gamma, \beta) = \sum_{i \in \llbracket N \rrbracket} \alpha_i c_i x^* \geq \sum_{i \in \llbracket N \rrbracket} \alpha_i \cdot c_i \cdot x^i = \sum_{i \in \llbracket N \rrbracket} \alpha_i g_i(\beta),$$

where the first equality follows Claim 2 of Lemma 9. Combined with the existence of $\alpha$ in Lemma 9, we derive that $h_A(\gamma, \beta) = \max_{\alpha \geq 0} \{ \sum_{q \in \llbracket Q \rrbracket} \alpha_q g_q(\beta) \mid \sum_{q \in \llbracket Q \rrbracket} \alpha_q c_q = \gamma \}$. $\qquad \square$

Since ($LP_\alpha(\gamma, \beta)$) is a linear program, for every $\gamma, \beta$, there must exists at least one basic feasible point that is an optimal solution. Hence, there exists a maximizer $\alpha$ of ($LP_\alpha(\gamma, \beta)$) that has at least $N - n$ elements equal to $0$ (non-basic variables), and at most $n$ non-zero elements (basic variables). The vectors of $\mathcal{C}$ corresponding to the basic variables generates a cone. Another way to view Lemma 10 is that, to find $h(\gamma, \beta)$, we need to find the cone containing $\gamma$ that maximizes $\sum_{i \in [\![N]\!]} \alpha_i g_i(\beta)$.

## A.4 Lemma 11

The next lemma shows that the optimal basis of ($LP_\alpha(\gamma, \beta)$) remains the same when $\gamma$ varies within the cone defined by that basis.

**Lemma 11.** *Let $b \in \mathcal{B}$ and $c \in \mathcal{C}$ be fixed. Without loss of generality, we assume that $c = \sum_{q=1}^{n} \alpha_q^* c_q$ where $\alpha_q^* > 0$ and the corresponding $C^* := [c_1, c_2, \ldots, c_n]$ is the optimal basis of $LP_\alpha(c, b)$. Then for any $\alpha_1, \ldots, \alpha_n \geq 0$ and $\gamma = \sum_{q=1}^{n} \alpha_q c_q$, the basis $C^*$ is also the optimal basis of $LP_\alpha(\gamma, b)$.*

*Proof.* By the hypothesis of the lemma and by Lemma 10, we have:

$$h_A(c, b) = \sum_{i=1}^{n} \alpha_i^* h_A(c_i, b).$$

Since $h_A(\gamma, b)$ is a concave piecewise linear function and $\gamma = \sum_{i=1}^{n} \alpha_i c_i$, we have

$$h_A(\gamma, b) = \sum_{q=1}^{n} \|\alpha\|_1 h_A \left( \sum \frac{\alpha_q}{\|\alpha\|_1} c_q, b \right) \geq \sum_{q=1}^{n} \alpha_q g_q(b).$$

We will show that $h(\gamma, b) = \sum_{q=1}^{n} \alpha_q g_q(b)$ for every $\gamma \in \text{cone}(C)$. Since $h_A$ is linear with respect to $\gamma$, we only need to prove that $h(\gamma, b) = \sum_{q=1}^{n} \lambda_q g_q(b)$ where $\gamma := \sum_{q=1}^{n} \lambda_q c_q \in \text{conv}(C)$. By contradiction, suppose that there exists $\gamma^* \in \text{argmax}_{\gamma \in \text{conv}(C)} h_A(\gamma, b)$ and $h_A(\gamma^*, b) > \sum_{q=1}^{n} \lambda_q^* c_q$. Let $c' := \frac{c}{\|\alpha^*\|_1}$ so that $c' \in \text{conv}(C)$. By contradiction hypothesis, we have $h_A(\gamma^*, b) > h_A(c', b)$. Let $d := \gamma^* - c'$, and consider the function $l(t) = h_A(c' + td, b)$. Since $l(t)$ is concave and $l(1) > l(t)$ for $t \in [0, 1)$, $l(t)$ is a linear function on $[0, 1]$. However, because $c'$ is an interior point of $\text{conv}(C)$, there exist $\epsilon > 0$ such that $c' - \epsilon d \in \text{conv}(C)$ and $l(-\epsilon) \leq l(0) - \epsilon(l(1) - l(0))$, which contradict the concavity of $h_A(\gamma, b)$. Therefore, we have

$$h(\gamma, b) = \sum_{q=1}^{n} \alpha_q g_q(b).$$

Moreover, because the optimal solution of $LP_\alpha(c_q, b)$ is $\alpha_q = 1$, $\alpha_j = 0 \; \forall j \neq q$, $C^*$ is also the optimal basis of $LP_\alpha(c_q, b)$. Thus, $C^*$ is the optimal basis $LP_\alpha(\gamma, b)$ for every $\alpha_i \geq 0, i \in [\![n]\!]$. $\square$

## A.5 Theorem 2

The previous Lemma 11 implies that for a fixed $b$ we can partition the finite set $C$ of Lemma 9 into subsets of size $n$, denoted $\mathcal{P}^b := \{C_1^b, \ldots, C_{v_b}^b\}$ such that

1. $\mathcal{C} = \cup_{v=1}^{v_b} \text{cone}(C_v^b)$, where the interiors of $\text{cone}(C_v^b)$ are mutually disjoint, and

2. If $\gamma \in \text{cone}(C_v^b)$, then $C_v^b$ is the optimal basis of $LP_\alpha(\gamma, b)$.

Since $C$ is finite, there can only be a finite number of such partitions. Thus, we can associate each $\beta \in \mathcal{B}$ with a partition of $C$ and group the vectors of $\mathcal{B}$ into groups of constraint bounds vectors with the same partition.

**GVF Representation Theorem:** *For a fixed matrix $A \in \mathbb{R}^{m \times n}$, there exists a set of $P$ piecewise linear functions $\{F_p : \mathbb{R}^n \to \mathbb{R}^Q\}_{p=1}^P$ and a piecewise linear convex function $G : \mathbb{R}^m \to \mathbb{R}^Q$ such that:*

$$h_A(\gamma, \beta) = \max_{p \in [\![P]\!]} \{F_p(\gamma)^T G(\beta)\} \; \forall \gamma \in \mathcal{C}, \beta \in \mathcal{B}.$$

*Proof.* We will construct the function $h_A(\gamma, \beta)$ as follows. Let $\{\mathcal{B}_1, \ldots, \mathcal{B}_P\}$ be the partition of $\mathcal{B}$ such that, for every $p \in [\![P]\!]$, $\mathcal{B}_p$ contains every constraint bounds vector $b$ with the same partition of $C$ from Lemma 11, i.e., $\mathcal{P}^{b^1} = \mathcal{P}^{b^2}$ if and only if $b^1, b^2 \in \mathcal{B}_k$ for some $k$. Note that $P$ is finite because $\mathcal{C}$ is finite. Furthermore, we denote $\mathcal{P}^p := \{C_1^p, \ldots, C_{v_p}^p\}$ to be the partition of $C$ associated with the set $\mathcal{B}_p$.

Note that $C_v^p$ is invertible since it comes from the basis of $LP_\alpha(\gamma, b_p)$. Let $F_p : \mathbb{R}^n \to \mathbb{R}^Q$ be such that

$$[F_p(\gamma)]_q = \begin{cases} [(C_v^p)^{-1}\gamma]_q & \text{if } c_q, \gamma \in C_v^p \\ 0 & \text{otherwise.} \end{cases}$$

In addition, let $G : \mathbb{R}^m \to \mathbb{R}^Q$ be defined as

$$G(\beta) = [g_1(\beta), g_2(\beta), \ldots, g_Q(\beta)],$$

which is convex piecewise linear since each $g_i$ is an LP value function. By our choice of $F_p$, for any $\gamma$, we have $F_{(p)}(\gamma) \geq 0$. Furthermore, let $\alpha = F_p(\gamma)$, we also have $\sum_{q=1}^{Q} c_q \alpha_q = \gamma$. Thus, $F_p(\gamma)$ is a feasible solution of $LP_\alpha(\gamma, \beta)$ for every $\gamma \in \mathcal{C}$, $\beta \in \mathcal{B}$, and $p \in [\![P]\!]$.

By Lemma 10, we have

$$h(\gamma, \beta) = \max_{p \in [\![P]\!]} \{F_p(\gamma)^T G(\beta)\}.$$

$\square$

# B  Complete Proofs of the GVF Unsupervised Learning Theory

## B.1  Proof of Theorem 3

**Theorem 3.** *Any function $\eta_A \in \mathcal{H}(A)$ has the following properties:*

1. *$\eta_A(\gamma, \cdot)$ is piecewise linear, convex, and monotonically decreasing for every fixed $\gamma \in \mathcal{C}$.*

2. *$\eta_A(\cdot, \beta)$ is piecewise linear for every fixed $\beta \in \mathcal{B}$.*

3. *$\eta_A(\gamma, \beta) \leq h_A(\gamma, \beta)$ for every fixed $\beta \in \mathcal{B}$ and $\gamma \in \mathcal{C}$.*

*Proof.* The properties of a function $\eta_A$ follows by the our choice for activation functions, weight-signed and output constraints.

1. $\eta_A(c, \cdot)$ is linear since the at every layer of the constraint bounds stack, we apply a linear transformation and a activation that is also pieceiwse linear. Since maximum or non-negative linear combination of convex functions is convex, $\eta_A(c, \cdot)$ is convex with respect to the constraints bound. In addition, the function is monotonically decreasing since the first layer of the constraint bounds stack has non-positive weights.

2. Similarly, because of linear transformations at every layer in the objective stack, the function $\eta_A(\cdot, b)$ with a fixed $b$ is piecewise linear.

3. For every fixed $c \in \mathcal{C}$, $\eta_A(c, \cdot)$ is a convex piecewise linear function. Therefore, $\eta_A(c, \beta)$ can be written as maximum of a finite number of linear function. Let $\eta_A(c, \beta) := \max_{l \in [\![L]\!]} \{\beta^T(y^l)\}$. Moreover, by the weight-singed constraint of the first of layer of the constraint bounds stack, we have $y^l \leq 0$. In addition, since $\eta_A \in \mathcal{H}(A)$, we have that $\max_{l \in [\![L]\!]} \{(a^i)^T(y^l)\} \leq c_i$ for every $i \in [\![n]\!]$. Therefore, each $y^l$ is a feasible solution of (2). By linear program weak duality, we have $\eta_A(c, \beta) \leq h_A(c, \beta)$. Since the argument applies for every $c \in \mathcal{C}$, we derive $\eta_A(\gamma, \beta) \leq h_A(\gamma, \beta)$ for every $\beta \in \mathcal{B}$ and $\gamma \in \mathcal{C}$.

$\square$

## B.2 Proof of Theorem 4

Property 3 of Theorem 3 relies the linear program weak duality. We will show that the inequality in Property 3 (Theorem 3) is indeed tight.

**Theorem 4**. *For any fixed $A \in \mathbb{R}^{m \times n}$, $h_A \in \mathcal{H}(A)$, and moreover $h_A$ is pointwise larger than all other elements of $\mathcal{H}(A)$.*

*Proof.* By the GVF Representation Theorem (Theorem 2), there exists continuous piecewise linear functions $\{F_i\}_{i=1}^p$ and convex piecewise linear function $G$ such that $h_A(\gamma, \beta) = \max_{i \in [\![p]\!]}\{F_i(\gamma)^T G(\beta)\}$. The objective stack can model exactly any continuous piecewise linear function. Therefore, with sufficient layers and neurons, it can model the functions $\{F_i\}_{i=1}^p$ exactly. Moreover, as shown in Theorem 2, each component of the function $G$ is a LPVF, and thus is a piecewise linear convex function with nonpositive slope. On the other hand, the constraint bounds stack with weight-sign constraint can model precisely any convex piecewise with nonpositive slope. Hence, with sufficient layers and neurons, it can also model the function $G$ exactly. Thus, $h_A \in \mathcal{H}(A)$. In addition, by Property 3 of Theorem 3, every function in $\mathcal{A}$ is upper bounded by $h_A$. □

## B.3 Proof of Theorem 5

**Theorem 5**. *There exists some $\mathcal{M} \in \mathbb{Z}_+^M$, some $\mathcal{N} \in \mathbb{Z}_+^N$, a finite set $\bar{\mathcal{C}} \subsetneq \mathcal{C}$, a finite set $\bar{\mathcal{B}} \subsetneq \mathcal{B}$, and some $\eta' \in \mathtt{DSM}(\mathcal{M}, \mathcal{N})$ such that $\eta'(\gamma, \beta) = h_A(\gamma, \beta)$ for all $\beta \in \bar{\mathcal{B}}, \gamma \in \bar{\mathcal{C}}$. Moreover, this same $\eta'$ necessarily satisfies $\eta'(\gamma, \beta) = h_A(\gamma, \beta)$ for all $\beta \in \mathcal{B}$ and $\gamma \in \mathcal{C}$.*

*Proof.* Let $C \in \mathcal{C} = \{c_1, \ldots, c_Q\}$ be the set describe in Lemma 9. Then for every cone generated by $n$ distinct vectors in $C$, we pick an arbitrary interior point, and denote the set containing every such interior points as $C'$. Then, we construct a finite set $\bar{\mathcal{C}} := C \cup C'$ as the union of $C$ and $C'$. We construct $\bar{\mathcal{B}}$ in a similar way. For every cone generated by $m$ distinct vectors in $\{a^1, \ldots, a^n, e^1, \ldots, e^m\}$, where $e^1, \ldots, e^m \in \mathbb{R}^m$ denotes the unit vector, we pick an arbitrary point, and denote the set containing every such interior points as $B'$. We then derive $\bar{\mathcal{B}} := \{a^1, \ldots, a^n, e^1, \ldots, e^m\} \cup B'$. The GVF Representation Theorem (Theorem 2) proves existence of functions $\{F_p\}_{p=1}^P$ and $G$ to model $h_A$. Since these functions are piecewise linear, we denote $f_p$ as the number of pieces of $F_p$ for $p \in [\![P]\!]$ and $g$ as number of pieces of $G$. We then pick features $\mathcal{N}$ such that the $\gamma$-stack of the DSM can represent every piecewise linear function with at most $f_1, \ldots, f_P$ pieces. Similarly, for the $\beta$-stack, we pick features $\mathcal{M}$ such that the stack can represent every convex piecewise linear function with at most $g$ pieces. [3]

Let $\eta' \in \mathtt{DSM}(\mathcal{M}, \mathcal{N})$ and $\eta'(\gamma, \beta) = h_A(\gamma, \beta) \ \forall \beta \in \bar{\mathcal{B}}, \gamma \in \bar{\mathcal{C}}$, we show that $\eta'(\gamma, \beta) = h_A(\gamma, \beta) \ \forall \beta \in \mathcal{B}, \gamma \in \mathcal{C}$. For a fixed $c \in \bar{\mathcal{C}}$, the function $\eta'(c, \beta)$, by Property 2 of Theorem 3, is a convex piecewise linear function. Moreover, $\eta'(c, \beta) = h_A(c, \beta)$ for every $\beta \in \bar{\mathcal{B}}$. By our choice of $\bar{\mathcal{B}}$, $\eta'(c, \beta) = h_A(c, \beta)$ at more point than the number of piece that $h_A(c, \beta)$ can have. Since $\eta'(c, \beta) \leq h_A(c, \beta) \ \forall \beta \in \mathcal{B}$ and $\eta'(c, \beta)$ is convex, we have $\eta'(c, \beta) = h_A(c, \beta)$. Hence, we have $\eta'(c_q, \beta) = g_q(\beta)$ for every $q \in [\![Q]\!], \beta \in \mathcal{B}$.

By Lemma 9, $h_A$ can be represented by a conic combination of $\{g_q(\beta)\}_{q=1}^Q$. By our choice $\bar{\mathcal{C}}$. The function $\eta'$ equal $h_A$ at more points than the number of pieces it can have. Thus $\eta'(\gamma, \beta) = h_A(\gamma, \beta)$ for every $\beta \in \mathcal{B}$ and $\gamma \in \mathcal{C}$. □

## B.4 Proof of Corollary 6 and 7

**Corollary 6** *Take the $\mathcal{M}, \mathcal{N}, \bar{\mathcal{B}},$ and $\bar{\mathcal{C}}$ that Theorem 5 guarantees must exist. Then, $h_A$ is the unique solution of* (7)

*Proof.* Theorem 5 implies $h_A \in \mathtt{DSM}(\mathcal{M}, \mathcal{N})$. By Theorem 3, we have that $h_A(c, \beta) \geq \eta(c, \beta)$ for every $\eta$ satisfies (7). Finally, Theorem 5 shows that we can represent $h_A$ through set of point $\bar{\mathcal{B}}$ and $\bar{\mathcal{C}}$. Therefore, $h_A$ is the optimal solution of (7). □

---

[3]Such features exist because we can construct a NN's architecture with one wide hidden layer.

**Corollary 7**. *Given any nonempty subsets $\mathcal{D}_b \in \mathcal{B}$ and $\mathcal{D}_c \in \mathcal{C}$, there exists a sufficient large $\mu$ such that $h_A$ is an optimal solution of* (8).

*Proof.* The Corollary follows from [40, Theorem 17.3] on local minimizer of nonlinear program with non-smooth penalty function and the fact that $h_A$ is the optimal solution of (7). □

## C   Uncapacitated Facility Location

*Uncapacitated facility location* (UFL) [54] is a classical, widely-applicable problem with the goal of deciding which of $n_f$ "facilities" to open (warehouses, plants, equipment, etc.) while taking into account a set of $n_c$ "customers" that are allocated to the open facilities. This can be interpreted as a deterministic two-stage problem: in the first stage, we select a subset of facility locations to open, and in the second stage, we assign each customer to a single open facility, while minimizing the costs of opening facilities plus customer allocation. Formally, the problem is as follows:

$$
\min_{x,y} \sum_{i \in [\![n_f]\!]} f_i y_i + \sum_{i \in [\![n_f]\!]} \sum_{j \in [\![n_c]\!]} c_{ij} x_{ij}
$$

$$
\sum_{i=1}^{n_f} x_{ij} = 1 \qquad\qquad \forall j \in [\![n_c]\!]
$$

$$
x_{ij} \le y_i \qquad\qquad \forall i \in [\![n_f]\!], j \in [\![n_c]\!]
$$

$$
x_{ij} \ge 0 \qquad\qquad \forall i \in [\![n_f]\!], j \in [\![n_c]\!]
$$

$$
y_i \in \{0,1\} \qquad\qquad \forall i \in [\![n_f]\!]
$$

where $n_f$ and $n_c$ are the number of facilities and customers respectively. Note that the second-stage variables are implied integer variables, i.e. if $y^*$ is integer, then there exists an optimal solution $(x^*, y^*)$ such that $x^*$ is integer. Therefore, we can effectively treat the second-stage variables $x$ as continuous variables.

We consider two sets of benchmark instances:

- *Euclidean*, in which we generate facilities and customers uniformly at random in a box $[0,1]^2$, using Euclidean distances as customer costs, with sizes $n_f = n_c \in \{100, 200, 300\}$; and

- *KG*, which are the symmetric Koerkel-Ghosh instances from the UFLLIB benchmark set [25], with sizes $n_f = n_c \in \{250, 500, 750\}$.

For Euclidean UFL instances, we generate one instance of Euclidean for training and five instances for testing. In the KG set from UFLLIB, we use first instance from the class A instances (a1) for training and test on the class B instances (classes A and B differ in how the costs were generated).

## D   Stochastic Capacitated Facility Location

The Stochastic Capacitated Facility Location (SCFL) problem is a variant of the traditional facility location problem where decision-making occurs under uncertainty regarding demand or other parameters, and facilities have limited capacities. SCFL problems are often modeled as two-stage stochastic programs. In the first stage, we make decisions involve selecting the facilities to open, committing to fixed setup costs. In the second stage, given the realized demand, the model assigns customers to open facilities, ensuring capacity constraints are respected while minimizing assignment costs. The problem can be formulated as follows:

$$
\min_{x,y} \sum_{i \in [\![n_f]\!]} f_i y_i + \sum_{s \in S} p_s \sum_{i \in [\![n_f]\!]} \sum_{j \in [\![n_c]\!]} c_{ij}^s x_{ij}^s
$$

$$
\sum_{i \in [\![n_f]\!]} x_{ij}^s \ge d_j^s, \qquad\qquad \forall j \in [\![n_c]\!], \forall s \in S
$$

$$\sum_{j \in [\![n_c]\!]} x_{ij}^s \le u_i y_i, \qquad\qquad \forall i \in [\![n_f]\!], \forall s \in S$$

$$x_{ij}^s \ge 0, \qquad\qquad \forall i \in [\![n_f]\!], \forall j \in [\![n_c]\!], \forall s \in S$$

$$y_i \in \{0,1\}, \qquad\qquad \forall i \in [\![n_f]\!],$$

where, similar to the UFL case, $n_f$ and $n_c$ denotes the number of facilities and customers, respectively. In addition, $S$ is the set of all scenarios and $p_s$ denote the probability of the scenario $s$. Moreover, we denote $f_i$ as the fixed cost of opening facility $i$, $c_{ij}^s$ as the assignment cost per unit from facility $i$ to customer $j$ in scenario $s$, $d_j^s$ as the demand of customer $j$ in scenario $s$, and $u_i$ as the capacity of facility $i$.

A major practical difference from UFL is that we can no longer decompose the problem over customers, due to capacity constraints that link all customers. However, this stochastic problem has scenarios that we decompose over, where both the demand and assignment costs can vary. Here, we have $n_c \cdot n_f$ variables in each subproblem, whereas in UFL we had $n_f$ variables per subproblem.

To generate SCFL instances, we take deterministic instances from the OR-Library [6], and generate $50$ scenarios by randomly modifying both the demands and unit costs to be $\mathcal{N}(\mu, \sigma)$ where $\mu$ is the original value and $\sigma$ is drawn from $\mathcal{U}(0.1\mu, 0.2\mu)$, resampling any negative costs. We consider all scenarios to be equally likely, i.e., $p_s = \frac{1}{|S|}$. We use $30$ samples for $\mathcal{D}_b$. We perform the experiment of learning the GVF function for SCFL in Table 3. We conduct the training on cap61 and testing on cap62, with 16 customers and 50 facilities.

Table 3: Comparison between Dual-Stack Model and DenseNet in Learning GVF.

| Class of GVF | | Dual-Stack Model | | | | DenseNet | | | Random Forest | |
| --- | --- | --- | --- | --- | --- | --- | --- | --- | --- | --- |
| | Train Time (s) | True Rel. Error | Train Lower Bound | Test Lower Bound | Data Label Time (s) | Train Time (s) | True Rel. Error | Train Time (s) | True Rel. Error |
| SCFL 16x50 | 294.89 | **6.75 %** | 93.46 % | 37.27 % | 40.21 | 280.44 | 10.69 % | 92.67 | 7.15 % |

While the description of UFL and SCFL are similar, the GVF of SCFL is more complex than UFL's. An important difference here is the feasibility of the right-hand side. For UFL, any vector in the unit-box domain is feasible. However, it is not the case for SCFL, e.g., when demand is greater than supply. Thus, in learning the GVF, we must take into account the possibility of infeasible right-hand side vectors. For the experiment in Table 3, we model the infeasible region by introducing right-hand side vector with $0$ supply and positive demand to the training data. The approximation of the value function is then used to solve the SCFL problem approximately.

Table 4: DSM Heuristic Solver on a1 SCFL Instance.

| Instances | | DSM Heuristic Solver | | | Full Model Solver | | Benders Heuristic |
| --- | --- | --- | --- | --- | --- | --- | --- |
| | Solve Time (s) | Provable Gap (%) | Gap to MILP (%) | Gap to Benders | LP Relaxation Solve Time (s) | MILP Solve Time (s) | Solve Time (s) |
| SCFL 16x50 | 14.09 | 43.05 | 2.29 | 1.25 | 0.5 | 64.64 | 5.78 |

# E    Details on Numerical Experiments

For comparison of the performance between DSM and DenseNet in learning GVF, we perform a parameter tuning for both models in terms of model architecture, dropout and learning rate. We train both of these models on a single GPU. With DSM, for the $\beta$-stack we perform training with $1, 2, 3$ layers, each with $32, 64$ neurons. The activation of every layer is max-pooling with a window of size 5. For the $\gamma$-stack, we also train with $2, 3$ layers each with $32, 64$ neurons, except the size of last layer is $128, 256$ respectively. In addition, to avoid vanishing gradients, we use a composition of GeLU and ReLU during training only. The final layer of $\gamma$-stack is then reshaped to $4 \times 32$ or $4 \times 64$ to match the shape for the output of $\beta$-stack. The dropout is only added in the last layer of the DSM and its parameter is chosen among $\{0.02, 0.03, 0.04\}$. With DenseNet, we also train the model with $1, 2, 3$ layers, each with $32, 64$ neurons. Dropout is added in every layer of the DenseNet and its parameter is chosen among $\{0.1, 0.2, 0.3\}$. The learning rate of Adam for both model is selected from $\{10^{-2}, 10^{-3}, 10^{-4}\}$. Table 5 summarizes the best configuration for each models.

Another aspect in learning GVF using our unsupervised framework is the choice of penalty $\mu$ update function (see Algorithm 1). We also want to answer how the quality of the upper bound at every training data point affect the learned function. As we can observe in Table 7 and 6 that the quality of

the learned function depends more on how we update the penalty term $\mu$ more than how good the upper bound is.

Table 5: NN Architectures and Learning Parameters for DSM and DenseNet

| Instances | | Dual-Stack Model | | | DenseNet | | |
| | | Model Arc. | Dropout | Learning Rate | Model Arc. | Dropout | Learning Rate |
|---|---|---|---|---|---|---|---|
| **KG** | 250 | $\mathcal{M} = [32], \mathcal{N} = [32, 128]$ | 0.03 | 1e-3 | [64, 64, 64] | 0.1 | 1e-3 |
| | 500 | $\mathcal{M} = [64], \mathcal{N} = [64, 256]$ | 0.03 | 1e-3 | [128, 128, 128] | 0.1 | 1e-3 |
| | 750 | $\mathcal{M} = [64], \mathcal{N} = [64, 256]$ | 0.03 | 1e-3 | [128, 128, 128] | 0.1 | 1e-3 |
| **Euclidean** | 100 | $\mathcal{M} = [32], \mathcal{N} = [32, 128]$ | 0.03 | 1e-2 | [64, 64, 64] | 0.1 | 1e-3 |
| | 200 | $\mathcal{M} = [64], \mathcal{N} = [64, 256]$ | 0.03 | 1e-2 | [128, 128, 128] | 0.1 | 1e-3 |
| | 300 | $\mathcal{M} = [64], \mathcal{N} = [64, 256]$ | 0.03 | 1e-2 | [128, 128, 128] | 0.1 | 1e-3 |

Table 6: Comparison of update methods and upper bounds for GVF Learning on KG-sym 250

| | upper bound = 2.0 | | upper bound = 100.0 | | upper bound = optimal value | | upper bound = optimal value x 2 | | unbound | |
| | #cons satisfied | true rel error | #cons satisfied | true rel error | #cons satisfied | rel abs loss | #cons satisfied | true rel error | #cons satisfied | true rel error |
|---|---|---|---|---|---|---|---|---|---|---|
| **linear update** | 93.02% | 5.32% | 68.00% | 2.73% | 100.00% | 21.53% | 100.00% | 10.04% | 92.06% | 5.94% |
| | 100.00% | 6.84% | 99.98% | 13.57% | 100.00% | 21.53% | 100.00% | 10.04% | 100.00% | 6.81% |
| **adaptive update** | 99.70% | 3.91% | 74.22% | 2.49% | 100.00% | 12.14% | 98.88% | 4.60% | 92.42% | 3.93% |
| | 100.00% | 4.02% | 99.63% | 4.89% | 100.00% | 12.14% | 100.00% | 5.14% | 99.58% | 4.15% |

Table 7: Comparison of update methods and upper bounds for GVF Learning on Euclidean 100

| | upper bound = 2.0 | | upper bound = 100.0 | | upper bound = optimal value | | upper bound = optimal value x 2 | | unbound | |
| | #cons satisfied | true rel error | #cons satisfied | true rel error | #cons satisfied | true rel error | #cons satisfied | true rel error | #cons satisfied | true rel error |
|---|---|---|---|---|---|---|---|---|---|---|
| **linear update** | 55.50% | 19.80% | 99.80% | 99.06% | 89.20% | 64.75% | 99.70% | 62.97% | 42.20% | 21.36% |
| | 87.30% | 65.43% | 99.80% | 99.06% | 100.00% | 82.81% | 100.00% | 65.43% | 97.70% | 81.79% |
| **adaptive update** | 51.00% | 23.23% | 62.40% | 20.27% | 100.00% | 63.02% | 98.10% | 48.39% | 49.30% | 25.75% |
| | 90.80% | 37.19% | 98.50% | 44.89% | 100.00% | 63.02% | 99.60% | 48.88% | 96.10% | 55.05% |

# F  Dual-Stack Model for Non-Standard LP Formulations

Table 8 describes the necessary changes to the DSM in the case that the LP is not in standard form. Alternatively, one may simply convert the LP to standard form.

Table 8: Alterations to the architecture for LPs not in standard form

| Linear Program Constraints | DSM Weight-Sign Constraints | Unsupervised Training Penalty Function |
|---|---|---|
| $Ax \leq b$ | Nonpositive first layer of $\beta$-Stack | None |
| $Ax \geq b$ | Nonnegative first layer of $\beta$-Stack | None |
| $Ax = b$ | No sign-constrained first layer of $\beta$-Stack | None |
| $x \in \mathbb{R}^n_+$ | None | $\mu \sum_{\gamma \in \mathcal{D}_c} \sum_{i \in [\![n]\!]} \cdot \max\{\eta^\theta_A(\gamma, a^i) - \gamma_i, 0\}$ |
| $x \leq \mathbb{R}^n_-$ | None | $\mu \sum_{\gamma \in \mathcal{D}_c} \sum_{i \in [\![n]\!]} \cdot \max\{\gamma_i - \eta^\theta_A(\gamma, a^i), 0\}$ |
| $x \in \mathbb{R}^n$ | None | $\mu \sum_{\gamma \in \mathcal{D}_c} \sum_{i \in [\![n]\!]} \cdot |\eta^\theta_A(\gamma, a^i) - \gamma_i|$ |

