# OpenReview forum: "Learning Generalized Linear Programming Value Functions"
_NeurIPS.cc/2024/Conference — NeurIPS 2024 spotlight_

### Official Review · Reviewer_qfDV · 2024-07-11

**Soundness:** 4
**Presentation:** 4
**Contribution:** 3
**Rating:** 7
**Confidence:** 5

**Summary:**

This article presents some contribution on the use of neural network to learn what is called the Generalized Linear Programming Value Function (GVF), that evaluates the optimal value of a linear program given as inputs the objective vector and the matrix constraints. This function is useful in many algorithms to actually find a solution (and not only the value) of the optimization problem, in particular for Mixed Integer Linear Programs (MILPs), or two-stage optimization problems. To do so, the authors first prove some fundamental decomposition of the GLV function. Then, they use this result to design a proper architecture of a neural network that can potentially compute that function. With additional analysis, they also provide a theoretical foundation for the use of unsupervised learning to compute the GVF. The authors then devise a heuristic based on these insights and demonstrate the efficiency of their approach experimentally.

The main two theoretical contributions are: i) a decomposition of the GVF into a max-product of two piecewise-linear functions (Theorem 2), and ii) a structural description (Theorem 2 and 3) of what a GVF must verify to be representable as some NN with the architecture they deduced from Theorem 2. In particular, ii) gives the existence of a finite set of parameters such that if the given architecture is equal to the GVF, then they must be equal everywhere.

The authors then present algorithms that incorporate heuristics, that they discuss in details, showing how they are grounded in theory.

**Strengths:**

The article is well-organized, reads well, and exposes clearly the contributions. It presents a serious study, with solid structural contributions. It can be of interest for researchers at the interface of optimization and machine learning, as well as developers of optimization solvers.

The authors also articulate well between these theoretical results and the practical algorithms.

**Weaknesses:**

- The experimental results are somehow limited. Although the double stack method (presented in this article) seems to be more effective than DenseNets in average, it is not always the case as shown by Table 1. Also, when compared to SCIP, the gaps for most of the instances are above 20% (although the running time is indeed less than a second).

- The authors do not provide an explanation of what they think to be the limitation of their experimental results, that would be of interest for readers who would be interested to follow up with their work. I would be willing to revise my evaluation score if this is adressed by the authors.

**Questions:**

- Is Theorem 3 completely new? Did it not exist before in the optimization literature?

- Where do you believe the bottleneck of the method resides (giving results of tables 1 and 2) ? Is it the difficulty to actually compute the GVF in an unsupervised manner?

- Typo: Line 90, the authors probably meant $\xi$, not $\chi$

**Limitations:**

The authors are transparent about the limitations of their work and results. Some additional explanations about the causes or bottleneck of these limitations would have been welcome. (see second question above).

---

> ### Author Rebuttal · Authors · 2024-08-07
>
> We appreciate the reviewer’s comments, especially for highlighting our contributions and the clarity of the paper. Please see the global rebuttal for expanded computational results. We address the specific comments (both from the Weaknesses and Questions section) below.
>
> 1. **Effectiveness of the method**: From our perspective, the major benefits of the DSM are that it guarantees that certain invariants that we know must hold in the “ground truth” function, and that these invariants can be translated effectively to efficient methods for our end goals. Producing models with lower errors than standard methods is a nice side effect of using a suitable architecture, but was not our main goal. In particular, we design the DSM such that: 1) the DSM can be easily embedded into an optimization problem as an LP due to its linearity and convexity on $\beta$, 2) it produces a function that is a guaranteed lower bound to the true GVF, and 3) it does not require supervised data. Due to these properties, we believe that our method is valuable even if our errors were the same, or even somewhat worse, than a reasonable baseline. Property 1 is what allows us the very fast solve times (<1 to ~4s) presented in Sec. 7.2, in problems where even solving the LP relaxation of the full model can take minutes. While this perspective may mean that our approach is not well-suited for every possible use case, there are settings where this approach is very well-suited: for example, soft real-time control where fast solve latencies is a hard requirement.
>
> 2. **Limitations + Bottleneck of the method**: This is a very good question, and we agree that a discussion on limitations can help researchers build upon our work. We enumerate potential limitations in each part of our method.
>
>    **(a) Stability during training**: During training, we must balance two terms in our loss function: one that rewards us for fitting the data well (first term), and another one that (softly) constrains the function down to be below the true value function for $\gamma \in \mathcal{D}_c$ (second term). Finding a stable balance between these two terms appears to be one of the most challenging parts of training. This was in fact a significant obstacle during the development of our method, and we overcome this limitation by proposing an adaptive update method for the penalty $\mu$, tuning the initial $\mu$ hyperparameter well, and proposing a good stopping criterion. However, even with all these measures, we do observe from the training logs that there is room for improvement (e.g., an adaptive stopping rule), which is a potential direction for future research.
>
>    **(b) Generalization to objective vectors not in the training set**: Our theoretical results say that guaranteeing the constraints (7b) is sufficient to produce a function that lower bounds the GVF, but we may have infinitely many constraints (as represented via $\bar{\mathcal{C}}$). In practice, we aim to enforce it for the objective vectors in the training data $\mathcal{D}_c$, which does give us this guarantee for any right-hand side $\beta$ and any objective vector from this training collection. We observe that with sufficient samples, we can generalize to other objective vectors, but we note that our Test Lower Bound column in Table 1 differs from the Train Lower Bound column by a significant amount. That said, we find that this still works well for purposes of a heuristic (see next item), but this could be a potential pitfall especially if one does not use enough training data.
>
>    **(c) Guaranteeing a lower bounding function of GVF**: As described in Sec. 5.3, we obtain a model that is close to satisfying the constraints (7b), and we then scale the function down to obtain a function that universally lower bounds the GVF. A potential pitfall here is that if the model is not close enough to satisfying the constraints, then we could be overly conservative in our scaling and end up with a function that is too far from GVF. In particular, we observed that this can happen if we do not stop the training at the right point. On the other hand, we also observe that for purposes of the heuristic in Sec. 6, obtaining an approximation with a good overall shape is often sufficient even if it is scaled down, since we care more about obtaining a solution close to optimal instead of an accurate representation of the objective values.
>
>    We will include a version of this discussion in the paper, and welcome any further feedback the reviewer might have. Specifically, we believe that the points discussed above are key when trying to identify the limitations of our method. In particular, the unsupervised training does require us to handle stability better during training. However, we can generally mitigate them with the efforts presented in our paper.
>
> 3. **Novelty of Theorem 3**: Theorem 3 is indeed new (to the best of our knowledge). The Dual-Stack Model is novel, and a characterization of its properties is also novel, though it is mostly based on well-established properties (e.g., the piecewise linearity of ReLU NNs). In particular, the DSM was designed to have the desirable properties from Theorem 3 in order to mimic our GVF representation theorem (Theorem 2).
>
> 4. **Typo**: Thank you for pointing out the typo. We will fix it in the final version.

---

> > ### Comment · Reviewer_qfDV · 2024-08-09
> >
> > Thank you for your feedback. I agree that a dedicated paragraph or subsection that sums up the bottlenecks and limitations would help greatly for follow-ups on this work.
> > Based on the answers of the reviewers I have decided to update my score to 7. I also believe that a more extensive set of experimental investigations and/or results would strengthen further the visibility and impact of this work.

---

> > > ### Author Response · Authors · 2024-08-13
> > >
> > > We agree that a discussion on limitations is helpful and we will add it, along with a more complete version of the additional experimental results discussed in the global rebuttal. Thank you for the review.

---

### Official Review · Reviewer_CMLg · 2024-07-12

**Soundness:** 4
**Presentation:** 4
**Contribution:** 4
**Rating:** 5
**Confidence:** 3

**Summary:**

This paper proposes a novel, theoretically grounded approach to learning values function (VF) of linear programs (LP).  Specifically, a Dual-Slack Model (DSM) is proposed to approximate the value function of linear programs with varying right-hand sides and objective coefficients.  The authors prove several key results used as motivation for the architecture and an unsupervised training procedure. This paper also presents numerical results that indicate their model can achieve similar and, in most cases, superior prediction of the LP value function. The authors also demonstrate how the DSM can be a heuristic for obtaining solutions to two-stage problems.

**Strengths:**

- The theoretically motivated architecture, i.e., the DSM, is an excellent contribution.  In my opinion, architecture and papers like this are a great contribution and will certainly be of significance to the field of learning-based approaches for optimization.  As such, I would rate the impact of the theory and model alone as significant.  The unsupervised learning approach is also a notable contribution.
- The prediction performance and solution quality of the model.  In prediction performance, i.e., learning the generalized value function, the DSM outperforms the dense network relatively consistently.  In addition, the DSM can compute high-quality first-stage solutions to two-stage MILPs.
- Quality and clarity of the paper.  Overall, the paper is very clearly written and well-motivated.  Additionally, the authors constantly discuss the limitations, which is undoubtedly important.

**Weaknesses:**

Overall, I have quite a favorable opinion of the paper.  However, a few weaknesses warrant further study, especially in the numerical experiments.  Additionally, while I briefly read over the proofs in the Appendix and did not have any concerns, I am certainly not an expert/familiar with some of the material.

- Limited computational results.  They propose a general approach yet only test six instances of the problem of uncapacitated facility location (UFL).   Moreover, the prediction performance compared to a feed-forward network model is not substantially better.  As such, comparing different benchmarks and perhaps even more models would be important to assess the computational performance holistically.  Even simple baselines, like gradient-boosted trees or random forests, would be useful, given these models generally predict with little to no parameter tuning.  Additionally, it appears no parameter tuning was done for DSM or the feed-forward, and given the relatively similar prediction and limited comparisons, it is hard to assess if DSM outperforms the feed-forward model substantially.  I agree that the paper's primary motivation and significance are more methodological/theoretical.  However, it would undoubtedly strengthen the empirical evidence for the approach proposed by the author to expand the computational section.
- The authors propose an unsupervised training approach that does not need to explicitly use the optimal objective values of LPs for training.  While this contribution may be helpful in some contexts, obtaining optimal objective values for LPs is generally computationally efficient.  Furthermore, solving these LPs can be trivially done in parallel, making the case for this even less clear.  From my understanding, the DSM model can also be trained with supervised learning, so comparing the trade-offs in time and solution quality between the supervised and unsupervised approach would be helpful.  A particular question that may be interesting to access is whether the supervised approach provides faster convergence in training such that it may outweigh the time required for data collection.  Additionally, perhaps the data requirements for the DSM would be significantly lower than those of a standard feed-forward network to represent the value function accurately, which may be an interesting result.
- Benchmarks and results for the heuristic two-stage problems.  The DSM model requires continuous variables in the second stage.  As such, it applies to all two-stage problems on which Benders decomposition can be used.  Given the large number of second-stage variables in the UFL instances, I suspect this may provide a more reasonable benchmark, especially when the LP times out, to access the solution quality of the DSM approach.  Additionally, the results as is demonstrate relatively high MILP gaps, except when the MILP does not solve the instance to optimality,

Given that the paper is relatively theoretically motivated but primarily would be valuable in applied contexts, strengthening the numerical experiments would undoubtedly improve the submission.

**Questions:**

- Was the data for the LPs solutions collected in parallel or sequentially?  If sequentially, it may be worth highlighting that parallel implementations can significantly reduce data collection time.
- What activation function $\sigma$ is used in the DSM model?
- How is (9) solved? The authors mention in line 273 that the model can written as an LP.  However, I believe it should be integer variables as it is piecewise linear.  If so, the authors should cite relevant literature, i.e., [1].
- How were the model sizes/parameters selected?
- Why did the authors choose not to provide code, especially given there is a computational study?  I see the authors mention that this is a theoretical paper in the checklist, but given they have an implementation, I see no reason to include it.

```
[1] Matteo Fischetti and Jason Jo. Deep neural networks and mixed integer linear optimization. Constraints,
23(3):296–309, 2018.
```

**Limitations:**

The methodology has limitations, such as applying generalized linear programming value functions. However, the authors explicitly address all limitations very clearly, which is certainly appreciated.

---

> ### Author Rebuttal · Authors · 2024-08-07
>
> Thank you for the detailed review. We are happy to hear that the reviewer appreciates the significance and clarity of the paper. We now will address each comment in both the Weaknesses and Questions section.
>
> 1. **Limited computational results**: As proposed, we will add both a new training baseline (Random Forests) and a new problem (SCFL). This is described in the global rebuttal and we hope that this strengthens the empirical evidence for our approach in your assessment. We also would like to highlight to the reviewer the following:
>    * Number of instances: We do not only test 6 instances of UFL, but rather we have 5 test instances for each of the 6 problem classes as described in Section 7. Therefore, we test 30 instances in total (45 with the addition of SCFL). We will make it clearer in the paper that each row in Tables 1 and 2 are averages over 5 instances.
>    * Hyperparameter tuning: We actually did perform parameter tuning for both DSM and DenseNet; please see Appendix D and Table 3. We will make this clearer in the final version of the text of the paper.
>
> 2. **(a) Training the DSM model with supervised data**: We did perform a comparison similar to the one proposed when we did an ablation test on the upper bound to use for the DSM. In particular, we set the upper bound to be the LP optimal value. The results are in Tables 4 and 5 in Appendix D (note: the first row is training and the second row is test; we missed these labels and will fix that in the final version). We observe that passing the true optimal value actually makes the relative error worse, in a way because of how we designed our loss function. Our intuition for why this is the case is as follows. In our loss function, we balance approximating the true value function well (which pushes the function up) and lower-bounding the value function via a penalty (which pushes the function down). If we use the actual optimal LP value in the approximation term, then the latter term is stronger, pushing the function down. Tables 4 and 5 suggest that in practice it is better to use an upper bound that is slightly above the true value function to balance out the penalty (which leads to our unsupervised approach). Finally, we emphasize that unsupervised vs supervised is not the only benefit of our approach: another key feature of the DSM is that it can be embedded in an optimization problem as an LP, rather than a MILP (see response to your question 3 below), which enables us to develop the very fast heuristic from Sec. 6.
>
>    **(b) Solving LPs can be trivially done in parallel**: This is of course correct, and we will add a mention of this in the paper. However, we would like to mention a few relevant points here. First, we note that the number of LPs that would have to be solved can be very large. In particular, we solve $n_f \cdot n_c / 10$ LPs in our computational setup. For example, for the KG case with 750 facilities and customers, this is 56,250 LPs. Second, in the case of UFL (and similarly for many large-scale problems amenable to decomposition approaches), the subproblem is very easy in that it can be actually solved with a greedy algorithm, which is what we do. Note that the average subproblem solve time is 187.02 / 56250 = 0.003s (3ms). For example, in the SCFL case (in the global rebuttal) with 50 scenarios and 30 samples for $\mathcal{D}_b$ (1,500 LPs), data labeling took 120s (i.e. 80ms per LP on average).
>
> 3. **Benchmarks and results for the heuristic two-stage problems**: We follow the reviewer’s suggestion and add a baseline where we use Benders’ decomposition within the heuristic framework from Sec. 6; see the global rebuttal for more details. We acknowledge that in certain cases we obtain relatively large gaps to the MILP solution, but 1) our approach performs very well on the KG class of instances, and 2) we believe that our approach can be of particular utility in practical applications where low latency is important, as we can achieve good solutions in the order of a few seconds or less.
>
> On the comments in the Questions section:
>
> 1. The data labeling process (i.e., solving LPs) was done sequentially. We will note in Section 7.1 that the data labeling time in Table 1 is sequential and can be embarrassingly parallelizable with sufficient computing power.
>
> 2. In our experiment, the activation function $\sigma$ for every layer of the $\gamma$ stack is ReLU. However, to avoid vanishing gradients, we use a composition of GeLU and ReLU for training only. With the $\beta$ stack, the activation of every layer is max-pooling with a window of size 5. The final layer of the DSM is a max-pool. We thank the reviewer for pointing this omission out; we intended for this discussion to be in Appendix D but missed its inclusion. It will be added to the updated version.
>
> 3. One of the main advantages of our approach is that the model can indeed be written as an LP. This is because we are learning a model that is piecewise linear convex on $\beta$. As described in Section 6, optimizing over input-convex neural networks does not require integer variables and can be done as an LP, and it is the reason why we see fast solve times in Table 2 (here, the first-stage variables are integer, but the DSM variables are continuous). We will make this clearer in the paper. In particular, we cited [3] earlier in the paper which studies input-convex NNs, and we will repeat the citation in Sec. 6.
>
> 4. The selection of model sizes and parameters is described in Appendix D. We do standard hyperparameter tuning by evaluating combinations of a small set of possible choices (see Appendix D for details on which values we tested).
>
> 5. Given our focus on the paper itself, our time at submission (and, now, for this rebuttal) was constrained and our code is not currently organized in such a way that it can be publicly released. That said, if the paper is accepted, we would be happy to publish the code for the model, method, and experiments.

---

> > ### Comment · Reviewer_CMLg · 2024-08-09
> > **Response to Rebuttal**
> >
> > Firstly, I would like to thank the authors for the rebuttal, for answering my questions, and for the additional computational results.  As most of my questions have been addressed, I will raise my score.
> >
> > At this point, the major reason that I am not raising my score even more is still the limited computational results.  Specifically, see the points below.
> >  - The computational results are still somewhat limited.  While the authors evaluated a new benchmark (SCFL).  However, it was very similar to UFL.  Generally, it would be ideal to compare to a benchmark less similar to UFL.
> >  - Improvements over the baseline (DenseNet) are still relatively marginal, and in some cases, such as the Euclidean UFL instances, DSM achieves quite poor solution quality.
> >  - As the authors highlight real-time control as a use case for their method, perhaps including the solution quality over time of DSM/MILP/Benders would be a nice addition.  For example, demonstrating that the solutions achieved by MILP/Benders at the time it takes the DSM to terminate would provide a stronger argument for DSM in this context.  Obviously, I do not expect this to be within the rebuttal timeframe, but perhaps in the final version of the paper.

---

> ### Author Response · Authors · 2024-08-10
>
> We appreciate your re-evaluation of our work. Thank you very much for spending the time reading our rebuttal and new experiments.
>
> To address some of your concerns, we would like to clarify some points:
> * You are absolutely right that the UFL and SCFL problems are similar, in terms of their standard formulation and the application that inspires them (and their names). However, there are nontrivial differences in the generalized value functions of the two problems that, we believe, mean that studying the two problems in the context of our work should offer some signal towards how generalizable the techniques we present are. First, we write down the subproblems (we use the notation from Appendix C for UFL, and in SCFL, $d$ and $s$ represent demands and capacities):
>    * UFL, for a fixed customer $j$:
>
>    $\\min \\{\\sum_j c_{ij} x_{ij}\\ |\\ \sum_i x_{ij} = 1,\\ x_{ij} \\leq y_i\\ \forall i \in [n_f]\\}$
>
>    * SCFL, for a fixed scenario $k$:
>
>    $\\min \\{\\sum_i \\sum_j c_{ijk} x_{ijk}\\ |\\ \sum_i x_{ijk} \geq d_{ik}\\ \forall j \in [n_c],\\ \\sum_j x_{ijk} \\leq s_i y_i\\ \forall i \in [n_f]\\}$
>
>    The main differences are the following:
>
>    1. An important difference here is the feasibility of the right-hand side. For UFL, any vector in the unit-box domain is feasible. However, it is not the case for SCFL, e.g., when demand is greater than supply. Thus, in learning the GVF, we must take into account the possibility of infeasible right-hand side vectors. This shows that DSM can be generalized to problems with non-trivial feasible regions as well.
>    2. In the SCFL, we not only vary the objective coefficients with each subproblem in the form of stochastic costs, but also the right-hand side in the form of stochastic demands (and the first-stage variables). In UFL, the variation of the right-hand side comes only from the first-stage variables.
>    3. In SCFL, we must handle both less-or-equal and greater-or-equal inequalities, whereas in UFL we only have less-or-equal constraints aside from the equality constraint.
>
> * While we are happy that our method shows improvement compared to DenseNet (even if marginally), the main contribution and advantage of our architecture and method are their **properties**, which enables us to:
>    1. Learn GVFs without the need for supervised data;
>    2. Embed the DSM efficiently into a larger optimization problem as LPs and not MILPs, which is what allows us to attain very fast runtimes from our heuristic;
>    3. Produce a function that is a guaranteed lower bound of GVFs.
>
>    Ultimately, the main objective of our work, from our perspective, is to lay the foundation for new ML-based decomposition techniques that can leverage the properties discussed above. We acknowledge that more research could be done to improve our method, but we stand by the value of our theoretical, methodological, and computational contribution as a whole, and hope that we can convince you of the same.
>
> * We definitely agree that adding plots on solution quality over time for the heuristics will make the picture clearer. We appreciate your suggestion and will add those plots in the final version of our paper.

---

### Official Review · Reviewer_bRXK · 2024-07-13

**Soundness:** 3
**Presentation:** 3
**Contribution:** 3
**Rating:** 6
**Confidence:** 3

**Summary:**

### Summary

Traditional LP Value Function (LPVF) represents the optimal value of a linear program as a function of its parameters, typically objective coefficients and constraint bounds. It is piecewise linear and convex, often used in sensitivity analysis and parametric programming. However, computing the LPVF can be computationally intensive and limited in capturing complex dependencies. The Generalized Value Function (GVF) introduced in this paper extends the concept of LPVF by modeling it as the maximum of bilinear functions, capturing more intricate interactions between parameters. And the approach is implemented through neural networks.

**Strengths:**

- Introduced a generalized value function framework that extends beyond traditional value functions.
- Proposed a neural network architecture tailored to this generalized value function.
- Developed a heuristic method based on the generalized value function for efficient optimization.

**Weaknesses:**

- Limited discussion on the potential advantages of GVF over LPVF, needing more thorough exploration and comparison.
- Experiments conducted on only one problem, lacking comparison with baselines from related studies.
- Experimental evaluation needs significant enhancement with more comprehensive testing and broader comparisons.

**Questions:**

- What are the detailed advantages and disadvantages of GVF compared to LPVF across various aspects?
- Why were other related studies not used as baselines in the experiments? Is there a justified reason for this omission?
- Why were the experiments conducted on only one problem? What is the rationale behind this limited scope?

**Limitations:**

This study does not pose any societal impact or ethical concerns. Other limitations have been addressed in the weaknesses and questions sections.

---

> ### Author Rebuttal · Authors · 2024-08-07
>
> Thank you for the review. Our new computational experiments, described in detail in the global rebuttal, expand our computational study with another family of problem instances and two new baselines (one for the model and one for the heuristic). We hope that this can address the reviewer’s concerns about the experiments.
>
> We answer each of the specific questions below:
> * **GVF vs LPVF**: The GVF extends the LPVF by allowing the objective coefficient to vary. In Sec. 2 of the paper, we write the following: “learning the entire GVF at once means that we can reuse the same learned model for many different objectives, potentially saving computation and allowing for a broader generalization.”, which is the main advantage of GVFs over LPVFs. While one can use multiple LPVFs to model subproblems in two-stage problems, it would require learning an NN for each LPVF. For example, in our UFL experiments, we would have to learn up to 750 NNs, whereas here we only learn a single one. Of course, learning a single GVF is generally harder than learning a single LPVF, though a core thesis of this work is that there is underlying structure tying together those many related LPVF that we should be able to exploit when learning the GVF.
> * **Other baselines**: We hope that this concern is alleviated with the addition of one more baseline for the model and another baseline for the heuristic, as described in the global rebuttal. Furthermore, the closest work on learning value functions is the Neur2SP paper (cited as [13] in our paper), which uses a feedforward ReLU network like in our DenseNet baseline. However, they take in as input a set of scenarios instead of a single one as in our case. Given all of this, we believe that DenseNet is a fair baseline, as it both closely hews towards the Neur2SP approach while adapting it in reasonable ways to our somewhat different setting. We also note in passing that the Neur2SP approach is not convex, and so its integration into the MILP heuristic we propose in Sec. 6 would require a MILP formulation (i.e., binary variables in the model), which would likely hinder its scalability.
> * **Other problems**: We hope that this issue is addressed with the addition of the SCFL problem as described in the global rebuttal.

---

> > ### Comment · Reviewer_bRXK · 2024-08-11
> >
> > Thank you for the comments. I think the rebuttal resolves many of my concerns. Please emphasize the strength of GVF compared to LPVF in more detail in the final version. I will raise my rating from 4 to 6.

---

> > > ### Author Response · Authors · 2024-08-13
> > >
> > > We will add more detail to the differences between GVF and LPVF in the final version. Thank you for your time reviewing our paper.

---

### Official Review · Reviewer_yPT2 · 2024-07-14

**Soundness:** 3
**Presentation:** 3
**Contribution:** 3
**Rating:** 6
**Confidence:** 3

**Summary:**

The paper presents a novel learning method for the Generalized Linear Programming Value Function (GVF), which models the optimal value of a linear programming (LP) problem as its objective and constraint bounds vary. The authors develop a neural network architecture, the Dual-Stack Model (DSM), that can efficiently approximate the GVF. This model is characterized by three properties: it provides a true under-approximation of the value function, is input-convex in the constraint bounds, and is trained using an unsupervised method that does not require computing LP optimal values. The paper also introduces SurrogateLIB, a library of MIP instances with embedded ML constraints, and demonstrates the effectiveness of the proposed method through computational experiments. Additionally, the authors develop a fast heuristic method for large-scale two-stage MILPs with continuous second-stage variables.

**Strengths:**

The paper provides a strong theoretical foundation for the GVF and its representation as a neural network, ensuring the model's structural properties align with the GVF. The Dual-Stack Model is a novel neural network architecture specifically designed for the GVF, which could inspire future work in similar areas. The method's ability to learn without the need for expensive LP solutions during training data generation is a significant advantage, especially for large-scale problems.

**Weaknesses:**

While the paper shows promising results, it is unclear how well the method generalizes to other types of optimization problems beyond the tested instances. There is limited comparison with existing methods for learning value functions or solving MILPs, making it difficult to assess the relative advantages of the proposed approach.

**Questions:**

How does the performance of the DSM scale with the size and complexity of the LP problems?
Can the unsupervised learning approach be extended to other types of value functions or optimization problems?

**Limitations:**

The paper does not provide open access to the code or data used in the experiments, which limits the ability of other researchers to reproduce and build upon the results.

---

> ### Author Rebuttal · Authors · 2024-08-07
>
> Thank you for the review. Please see the global rebuttal for our response on the limitation of the computational results raised in the “Weaknesses” section. We hope that it addresses your concerns regarding how our approach generalizes to other types of optimization problems, and regarding how it compares with other existing methods.
>
> Regarding the questions:
>
> * **Scalability**: We study the scalability of our method in Table 1, in which we vary facility location sizes from 250 to 750 for the KG instances and from 100 to 300 for the Euclidean instances. We highlight that these larger instances can be quite difficult: e.g., the LP relaxations of KG 500 and 750 take 6 min. and >15 min., respectively, to solve without decomposition. While training time of course increases with instance size, both the True Relative Error and Lower Bounds do not deteriorate in the case of KG and increase slightly in the case of Euclidean, which is evidence that DSM scales well. We will add a brief discussion on scalability in the paper.
> * **Extending the unsupervised learning approach**: Our unsupervised learning method is tightly linked with the theoretical structure of the GVF, in particular via Corollary 6. Therefore, applying this to other value functions would require investigating their structure, which is somewhat orthogonal. For example, an immediate direction could be to build an architecture for MILP value functions based on its subadditive properties. This would be more challenging, but we hope that our work could inspire further research in this direction.
>
> The reviewer also mentions access to code as a limitation. We would happily publish the code for the model, method, and experiments upon acceptance.

---

### Author Rebuttal · Authors · 2024-08-07

We thank all the reviewers for their feedback. We wrote this paper with the goal of presenting a “theoretically motivated architecture” for what we view as an important and interesting setting in mathematical optimization, and we are happy to see that this vision appears to have resonated with the review team. We are grateful that the reviewers view the paper as “well-organized” and “very clearly written” (we spent a lot of time on this!). We also appreciate the constructive feedback that they offered, which we feel will lead to a stronger and more compelling paper. We will spend the remainder of the rebuttal focusing on this feedback, and how we plan to use it to improve the paper. In particular, we have expanded the computational experiments to address reviewer concerns on limited experiments:

1. **New model baseline: Random Forests**. We have performed a comparison with the GVF learned with Random Forests (suggested by reviewer CMLg) in the attached pdf Table 1, which we will add to the paper. We used the sklearn package with 100 estimators and 8 parallel threads, and all other settings are set to default. We will also add random forest experiments for KG 750 (which we did not have time to perform during the rebuttal period) and the new problem class SCFL (see below) in the final version.

   As a reminder, these results should be interpreted in view of the three beneficial properties of our method: 1, the DSM can be easily embedded into an optimization problem as an LP due to its linearity and convexity on $\beta$, 2, it produces a function that is a guaranteed lower bound to the true GVF, and 3, it does not require supervised data. Random Forests do not benefit from any of these properties. It could be embedded into optimization as an MILP, which can make for example the heuristic in Sec. 7.2 much more expensive. Furthermore, they are piecewise constant and discontinuous, therefore not matching the natural structure of the GVF. Hence, even though Random Forests can attain a slightly lower error in some instances, this is still a positive result for the DSM especially given the long Random Forest training times.

2. **New heuristic baseline: MILP heuristic with Benders decomposition**. We compare our heuristic with a Benders Decomposition baseline (suggested by reviewer CMLg) similar to our Algorithm 3, except that in (9) we replace the DSMs by approximations obtained by solving the LP relaxation with Benders with t iterations, where t equals number of facilities. The results are in the attached pdf Table 2. After that, we solve (9) with a time limit of 1 minute (though the Euclidean instances solve to optimality faster). The solve times in the table are end-to-end. We observe that our heuristic performs much better than the Benders baseline for the KG instances, though Benders outperforms our heuristic for the Euclidean instances.

3. **New problem: Stochastic Capacitated Facility Location**. We performed experiments on the Stochastic Capacitated Facility Location Problem in the attached pdf Table 3. This is similar to the Uncapacitated Facility Location Problem in the paper, with two differences: 1, capacity constraints prevent us from decomposing over customers (facilities can serve a limited capacity over all customers), and 2, instead of a deterministic problem, we have multiple scenarios with varying demand and costs, and we would like to minimize the total expected cost over the scenarios. In this case, the decomposition is done over scenarios, and each subproblem in this decomposition is an assignment problem from customers to a set of fixed open facilities (decided in the first stage), satisfying required customer demands and facility capacity limits.

   To generate these instances, we take deterministic instances from [OR-Library](https://people.brunel.ac.uk/~mastjjb/jeb/info.html), and generate 50 scenarios by randomly modified both the demands and unit costs to be $\mathcal{N}(\mu, \sigma)$ where $\mu$ is the original value and $\sigma$ is drawn from $\mathcal{U}(0.1 \mu, 0.3 \mu)$. We use 30 samples for $\mathcal{D}_b$. Due to time constraints of the rebuttal, we only show one row of the table with one test instance (training on cap61 and testing on cap62, with 16 customers and 50 facilities). We will perform the full experiments for the final version, including the MILP heuristic.

   Unfortunately, our stopping criterion does not work as well in this case. We choose to stop at a fixed number of iterations, T = 60 (with 100 steps of the Adam algorithm in between), with the acknowledgement that training can be unstable. We decide to present these results in the rebuttal anyway because we believe they are still interesting, and we will further tune the training for the final version of the paper.

All other questions are addressed in the individual rebuttals. We are happy to discuss and incorporate further feedback.

---

### Decision · Program_Chairs · 2024-09-25

**Decision:**

Accept (spotlight)

**Comment:**

Good paper. Clear accept.

The paper studies the learning of the optimal objective function of a linear program as a function of its inputs. The neural network design utilizes the LP's optimality condition (and its implication that the optimal value function should be piecewise-linear one). The proposed method enjoys theoretical guarantees and gives promising numerical performances. And I agree with the reviewers that the paper can inspire many follow-up works on this topic.